# Antiviral type III CRISPR signalling via conjugation of ATP and SAM

Haotian Chi[1], Ville Hoikkala[1,2], Sabine Grüschow[1], Shirley Graham[1], Sally Shirran[1] & Malcolm F. White[1✉]

CRISPR systems are widespread in the prokaryotic world, providing adaptive immunity against mobile genetic elements[1,2]. Type III CRISPR systems, with the signature gene *cas10*, use CRISPR RNA to detect non-self RNA, activating the enzymatic Cas10 subunit to defend the cell against mobile genetic elements either directly, via the integral histidine–aspartate (HD) nuclease domain[3–5] or indirectly, via synthesis of cyclic oligoadenylate second messengers to activate diverse ancillary effectors[6–9]. A subset of type III CRISPR systems encode an uncharacterized CorA-family membrane protein and an associated NrN family phosphodiesterase that are predicted to function in antiviral defence. Here we demonstrate that the CorA-associated type III-B (Cmr) CRISPR system from *Bacteroides fragilis* provides immunity against mobile genetic elements when expressed in *Escherichia coli*. However, *B. fragilis* Cmr does not synthesize cyclic oligoadenylate species on activation, instead generating *S*-adenosyl methionine (SAM)-AMP (SAM is also known as AdoMet) by conjugating ATP to SAM via a phosphodiester bond. Once synthesized, SAM-AMP binds to the CorA effector, presumably leading to cell dormancy or death by disruption of the membrane integrity. SAM-AMP is degraded by CRISPR-associated phosphodiesterases or a SAM-AMP lyase, potentially providing an 'off switch' analogous to cyclic oligoadenylate-specific ring nucleases[10]. SAM-AMP thus represents a new class of second messenger for antiviral signalling, which may function in different roles in diverse cellular contexts.

*Bacteroides* spp. are Gram-negative, anaerobic bacteria that constitute a significant portion of the human gut microbiome[11]. The *Bacteroidales* are host to the most widespread and abundant phage found in the human digestive system, CrAssphage[12]. *B. fragilis* is an opportunistic pathogen, and is responsible for more than 70% of *Bacteroides* infections[13]. Bioinformatic analyses have revealed the presence of three CRISPR types—I-B, II-C and III-B—in *B. fragilis* strains, with the type III-B system being the most common[14]. Sequence analysis shows that *B. fragilis* Cas10, the main enzymatic subunit of type III effectors, lacks an HD nuclease domain but has an intact cyclase domain, similar to the *Vibrio metoecus* Cas10[15]. This suggests that the system functions via cyclic oligoadenylate (cOA) signalling to associated ancillary effectors. In *B. fragilis* and more generally in the *Cytophaga–Bacteroides–Flavobacterium* bacterial phylum these type III CRISPR systems are strongly associated with an uncharacterized gene encoding a divergent member of the CorA-family of divalent cation channel proteins[16,17] (Fig. 1a). The CRISPR-associated CorA proteins have not been studied biochemically but are predicted to consist of a C-terminal membrane spanning helical domain fused to a larger N-terminal domain with a unique fold. To investigate this further, we first generated a phylogenetic tree of Cas10 proteins and identified those associated with a gene encoding the CorA protein. Three phylogenetically distinct clusters of CorA-associated type III CRISPR systems were apparent, with the largest (CorA-1) being associated with type III-B systems (Fig. 1a).

The genomic context of CorA-containing type III CRISPR loci from cluster CorA-1 (Fig. 1b) reveals that the *corA* gene is typically found next to a gene encoding a phosphodiesterase (PDE)—the DHH-family nuclease NrN in the case of *B. fragilis* and *M. vanielii*, and a DEDD-family nuclease in the case of *S. oralis* and *S. lipocalidus*. In the genome of *A. butzleri* and related species, the *nrn* and *corA* genes are fused, suggestive of a close functional relationship. The closest predicted structural matches for *B. fragilis* NrN are to the pGpG-specific PDE PggH from *Vibrio cholerae*, which has a role in the turnover of the cyclic nucleotide c-di-GMP[18] and the GdpP PDE from *Staphylococcus aureus*, which degrades pApA molecules as a component of c-di-AMP signalling systems[19]. Analysis of the DEDD protein suggests structural matches to RNaseT[20], oligoribonuclease[21], and the mammalian REXO2 protein, which degrades linear RNA and DNA dinucleotides[22]. Thus, the CRISPR-associated NrN and DEDD proteins appear to be homologous to protein families that degrade small RNA and DNA species. Of note, in some CorA-containing type III systems including *Clostridium botulinum*, the PDE is replaced by a protein that is predicted to resemble a family of phage SAM lyase enzymes involved in evasion of host immune systems[23,24] (Fig. 1b).

[1]Biomedical Sciences Research Complex, School of Biology, University of St Andrews, St Andrews, UK. [2]University of Jyväskylä, Department of Biological and Environmental Science and Nanoscience Center, Jyväskylä, Finland. ✉e-mail: mfw2@st-andrews.ac.uk

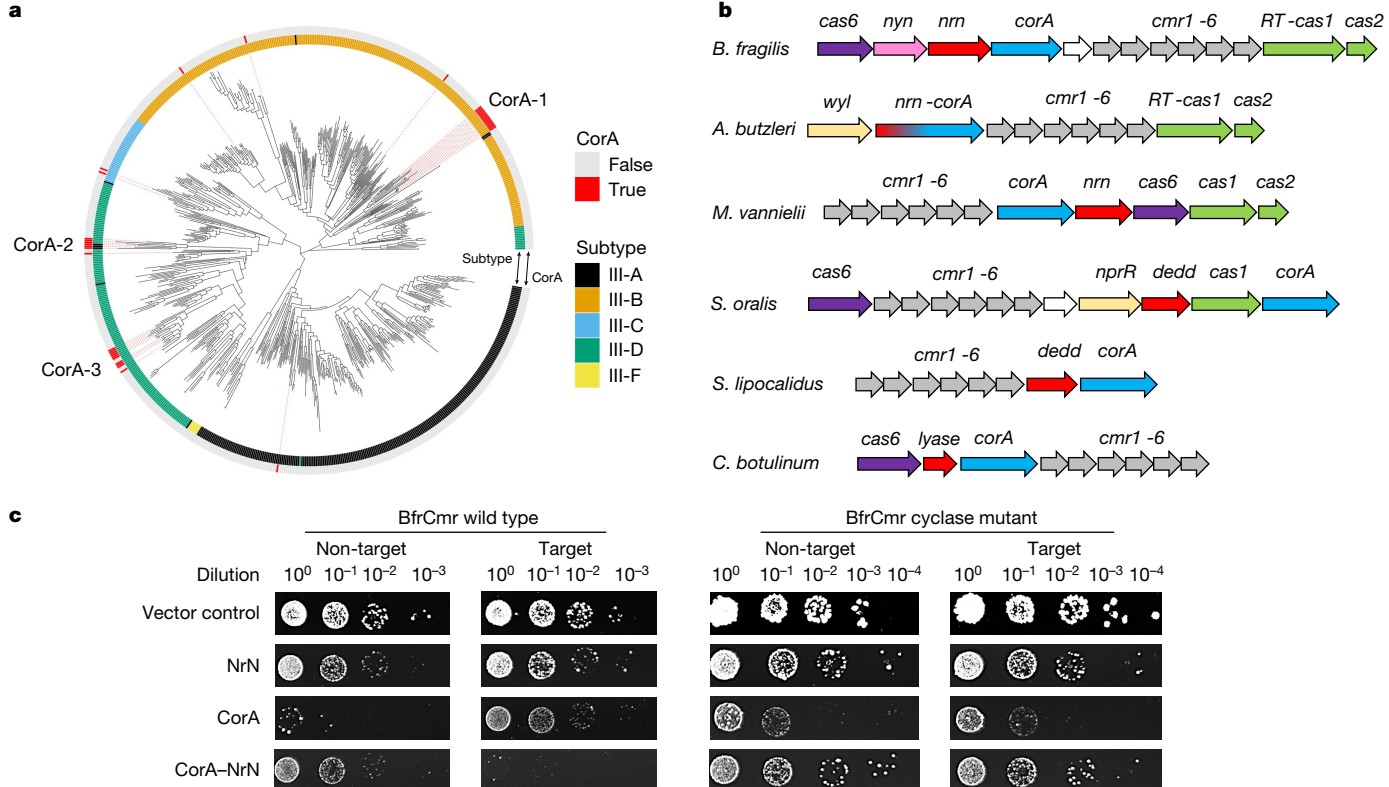

**Fig. 1 | Type III CRISPR systems with a CorA effector. a**, A phylogenetic tree of Cas10 proteins from type III CRISPR systems of complete bacterial and archaeal genomes, colour coded by subtype[42]. Red bars on the outer ring indicate systems associated with a CorA-family effector protein. There are three main clusters of CorA-associated Cas10s, labelled CorA-1, CorA-2 and CorA-3. **b**, Genome context and effectors of selected type III-B CRISPR systems with a *corA* gene (cluster CorA-1: *B. fragilis, Aliarcobacter butzleri, Methanococcus vannielii, Streptococcus oralis, Snytrophothermus lipocalidus, Clostridium botulinum*). The type III-B *cas* genes *cmr1–6* are shown in grey, with *cas6* in purple and the adaptation genes *cas1* (or a gene encoding a fused reverse transcriptase–Cas1 protein) and *cas2* in green. The putative membrane channel protein is encoded by the *corA* gene

(blue), which is adjacent to or fused with the genes encoding the PDEs NrN or DEDD (red). In *C. botulinum*, the PDE is replaced with a predicted SAM lyase. The *wyl* and *nprR* genes encode predicted transcriptional regulators. **c**, Plasmid challenge assay. *E. coli* BL21 Star cells expressing *B. fragilis* Cmr (wild-type or cyclase-defective variant) programmed with target (tetR) or non-target (pUC19) CRISPR RNA (crRNA) species were transformed with a pRAT plasmid that expressed the NrN and/or CorA proteins and carried a tetracycline resistance gene. Resistance was observed only when a targeting crRNA, active cyclase and both effector proteins were all present. Raw data are presented in Supplementary Data Fig. 1.

## *B. fragilis* Cmr is active in vivo

To investigate the activity of the *B. fragilis* type III CRISPR system, two plasmids were constructed. Plasmid pBfrCmr1-6, built using Gibson assembly[25], expresses synthetic versions of the codon-optimized genes *cmr1–6* and plasmid pBfrCRISPR encodes Cas6 and a mini-CRISPR array (Extended Data Fig. 1a). We expressed the complex in *E. coli* with a targeting (pCRISPR-Tet) or non-targeting (pCRISPR-pUC) crRNA and challenged cells by transformation with a pRAT-Duet plasmid expressing one or both of the CorA and NrN effector proteins (Fig. 1c). The pRAT-Duet plasmid also contains the *tetR* gene for activation of *B. fragilis* Cmr carrying the targeting crRNA. Cells were transformed with the pRAT-Duet vectors and grown in the presence of tetracycline to select for transformants. We included vectors expressing wild-type and cyclase-defective (Cas10 D328A/D329A variant) Cmr for comparison. We previously used this experimental design to investigate the *V. metoecus* Cmr system[15]. In conditions in which the Cmr system was activated and had the required ancillary effector proteins, lower numbers of colony-forming units were expected. The vector control (no effectors) served as a baseline for transformation. When only the NrN effector was present, no reduction in colonies was observed, suggesting no active targeting. When only the CorA protein was expressed, fewer colonies were observed in both target and non-target conditions, for both the wild-type and cyclase-deficient mutant (Δcyclase) Cmr,

suggesting some toxicity of the CorA protein. When both the CorA and NrN effectors were expressed, immunity—indicated by a markedly reduced number of colonies—was observed only for the wild-type Cmr system with *tetR* targeting. Immunity was lost when wild-type NrN was substituted with a variant mutated in the DHH active site motif (D85A/H86A/H87A), or when the CorA protein was truncated to remove the transmembrane domain (Extended Data Fig. 1b).

These data suggest that the *B. fragilis* Cmr system is functional in *E. coli* and requires the activity of the Cas10 cyclase domain and the presence of both effector proteins. The toxicity of CorA appears to be reduced by the presence of the NrN protein, regardless of activation of the type III system, suggestive of a strong functional link. Intriguingly, the type III-B complex from *Mycobacterium tuberculosis*, which synthesizes cyclic oligoadenylate 3–6 ($cA_{3–6}$) in vitro[26], did not provide immunity when combined with the NrN and CorA effectors, hinting at a non-canonical activation mechanism (Extended Data Fig. 1c). The strict requirement for the SAM-AMP degrading NrN protein in addition to CorA for plasmid immunity is an unusual aspect of the system and is discussed further below.

## RNA processing and degradation

Co-transformation of the expression plasmids into *E. coli* strain BL21 (DE3) enabled the expression of the *B. fragilis* Cmr effector and

purification by immobilized metal-affinity and size-exclusion chromatography (Extended Data Fig. 2a). We also purified *B. fragilis* Cas6 individually using the same chromatography steps. We first confirmed that Cas6 processed crRNA in the expected manner. The recombinant Cas6 enzyme cleaved synthetic fluorescein (FAM)-labelled crRNA at the base of a predicted hairpin with a 2-bp stem, reminiscent of *Methanococcus maripaludis* Cas6b[27]. This generates a canonical 8-nucleotide (nt) 5′ handle, (Extended Data Fig. 2b). Cleavage of an in vitro transcript comprising 2 repeats flanking one spacer generated the expected set of reaction products, culminating in a processed crRNA of 72 nt (Extended Data Fig. 2c). To investigate the composition of the crRNA present in the effector complex purified from *E. coli*, we isolated and labelled the crRNA using γ-$^{32}$P-ATP and polynucleotide kinase. This revealed 3 major crRNA species differing in length by 6 nt (Extended Data Fig. 3a,b). These products correspond to 3′ end trimming of the crRNA to remove the repeat-derived sequence and probably reflect effector complexes that differ in the number of Cas7 subunits and thus length of backbone, as has been seen for other type III systems (reviewed in ref. 28).

Type III CRISPR systems also cleave bound target RNA using the Cas7 subunit, either for direct defence against mobile genetic elements[29] or for regulatory purposes[8]. We proceeded to test for cleavage of target RNA bound to the crRNA in the effector. The 5′-end-labelled target RNA was cleaved at 4 positions with 6-nt spacing, corresponding to the placement of the Cas7 active sites in the backbone[29]. Cleavage was extremely rapid and was essentially complete after 2 min, the first time point (Extended Data Fig. 3c). As these sites interconvert and site 1 is furthest from the 5′ label, this cleavage was only observed for the Cmr4 or Cas7 D27A variant, which cleaves target RNA more slowly (Extended Data Fig. 3d). We also observed cleavage of target RNA at the boundary of the crRNA–target RNA duplex. This activity, which has not been observed for other type III systems, appeared to be due to the Cmr4 subunit, as it was not observed for the D27A variant. As target RNA cleavage has been shown to correlate with the deactivation of the Cas10 subunit[8,30], this suggests that the Cmr complex remains active for a very short time after target RNA binding. This groups *B. fragilis* Cmr together with the type III effectors from *Streptococcus thermophilus* and *Thermotoga maritima*, which cleave target RNA rapidly[5,30]. By contrast, the type III systems from *S. solfataricus* and *V. metoecus* have much slower RNA cleavage kinetics[8,15]. In the absence of structural data to define the number and positions of Cas7 subunits in the complex, we could not analyse the cleavage pattern further with any degree of certainty.

## Identification of the signalling molecule

As *B. fragilis* Cmr lacks a HD nuclease domain in the Cas10 subunit, immune function would be expected to be mediated by the cyclase domain via the generation of nucleotide second messengers. However, although the system provided cyclase-dependent immunity in vivo, activation of the wild-type Cmr in vitro resulted in very low yields of any observable product when incubated with ATP, in contrast to the Cmr complex from *V. metoecus*, which synthesizes cA$_3$[15]. This hinted at the possibility that a vital component was missing in the in vitro assays. Accordingly, we activated *B. fragilis* Cmr in *E. coli* using a plasmid to express target RNA and then processed cell lysates to allow isolation of nucleotide products. These were purified and analysed by high-performance liquid chromatography (HPLC). A prominent peak was observed following HPLC of extracts with activated Cmr, which was absent in the absence of target RNA or when the cyclase activity was knocked out by mutagenesis (ΔCy) (Fig. 2a). Mass spectrometry yielded a *m/z* value of 728.196 for the positive ion; to our knowledge, this *m/z* value did not correspond with any known cyclic nucleotide or indeed any other previously characterized metabolite (Fig. 2b). To identify the product, we fragmented the purified molecule using

tandem mass spectrometry (MS/MS). This enabled the identification of fragments characteristic of AMP and methionine (Fig. 2c). Further examination suggested that the molecule under study was SAM that was adenylated on the ribose moiety (Fig. 2d), a molecule that we hereafter refer to as SAM-AMP. To our knowledge, SAM-AMP has not previously been described in the literature—from either chemical or enzymatic synthesis perspectives—suggesting that it is a previously undiscovered class of signalling molecule.

To confirm that *B. fragilis* Cmr synthesized SAM-AMP, we reconstituted the reaction in vitro with ATP and SAM, analysing reaction products by HPLC and thin-layer chromatography (TLC) (Fig. 2e,f). We observed SAM-AMP production when SAM and ATP were present in vitro. *S*-adenosyl-ʟ-homocysteine (SAH) and the SAM analogue sinefungin[31], which differ at the sulfur centre, were also conjugated with ATP by Cmr (Fig. 2e,f and Extended Data Fig. 4c). No significant products were observed in the presence of ATP or all four ribonucleotides. The synthesis of SAM-AMP and SAH-AMP by *B. fragilis* Cmr were consistent with rapid, multiple-turnover kinetics that were essentially complete within the first 2 min of the reaction (Extended Data Fig. 4a,b). The observation of only SAM-AMP, and not SAH-AMP, in *E. coli* cell extracts is probably the result of the much higher concentration of SAM than SAH in *E. coli*[32] (0.4 mM versus 1.3 µM). Overall, these data provide strong evidence that the *B. fragilis* Cmr system generates a previously undescribed conjugate of SAM and ATP, rather than cOA.

Since Cas10 family enzymes synthesize 3′–5′ phosphodiester bonds[6,7], we considered it likely that SAM was fused to AMP at the 3′ position on the ribose ring, but the mass spectrometry data did not rule out a 2′–5′ phosphodiester bond. To address this, we incubated SAM-AMP and SAH-AMP with nuclease P1, which is specific for 3′–5′ phosphodiester bonds. Whereas SAH-AMP was completely degraded by nuclease P1, we observed only partial degradation of SAM-AMP (Extended Data Fig. 5). Thus, although we consider a 3′–5′ phosphodiester linkage to be likely, we cannot rule out a 2′–5′ linkage completely. Final confirmation of the linkage will require further analysis—for example, by NMR.

The crystal structure of *Pyrococcus furiosus* Cas10–Cas5 bound to two ATP molecules[33] shows one ATP in the 'donor' ATP1 site next to the GGDD cyclase catalytic motif and another in the 'acceptor' site (Fig. 3a). For the enzymes that synthesize SAM-AMP, SAM must bind in the acceptor ATP2 binding site, next to ATP1 in the donor site[34]. This arrangement would allow nucleophilic attack from the 3′-hydroxyl group of SAM to the α-phosphate of ATP1, resulting in the formation of a 3′–5′ phosphodiester bond linking SAM with AMP, and the release of pyrophosphate. The reaction chemistry is essentially the same as the one that takes place in canonical type III CRISPR systems that synthesize cOA species[34]. The major difference is that the triphosphate of ATP2 in the acceptor site is replaced by the methionine moiety of SAM, resulting in a change in local charge of the ligand from net negative to net positive, raising the possibility that Cas10s binding SAM will have a less basic binding site in this area. Examination of sequence conservation in the Cas10s associated with a CorA-1 cluster (Extended Data Fig. 6) revealed the presence of two absolutely conserved acidic residues, D70 and E151. Modelling of the *B. fragilis* Cas10 structure places these two residues in the vicinity of the methionine moiety of SAM in the acceptor site (Fig. 3b). D70 occupies the position equivalent to N300 in *P. furiosus* Cas10, which neighbours the β-phosphate of the acceptor ATP2 ligand, while E151 is in a similar position to R436, which forms a bidentate hydrogen bond with the γ-phosphate (Fig. 3a). We created variants of Cas10 with D70N, E151R and D70N/E151R mutations, which were expressed and purified as for the wild-type protein (Extended Data Fig. 2a), and assessed them for their ability to synthesize nucleotide products (Fig. 3c). The E151R variant had a limited effect on SAM-AMP synthase activity, but the D70N variant was significantly compromised and the double mutant showed no detectable

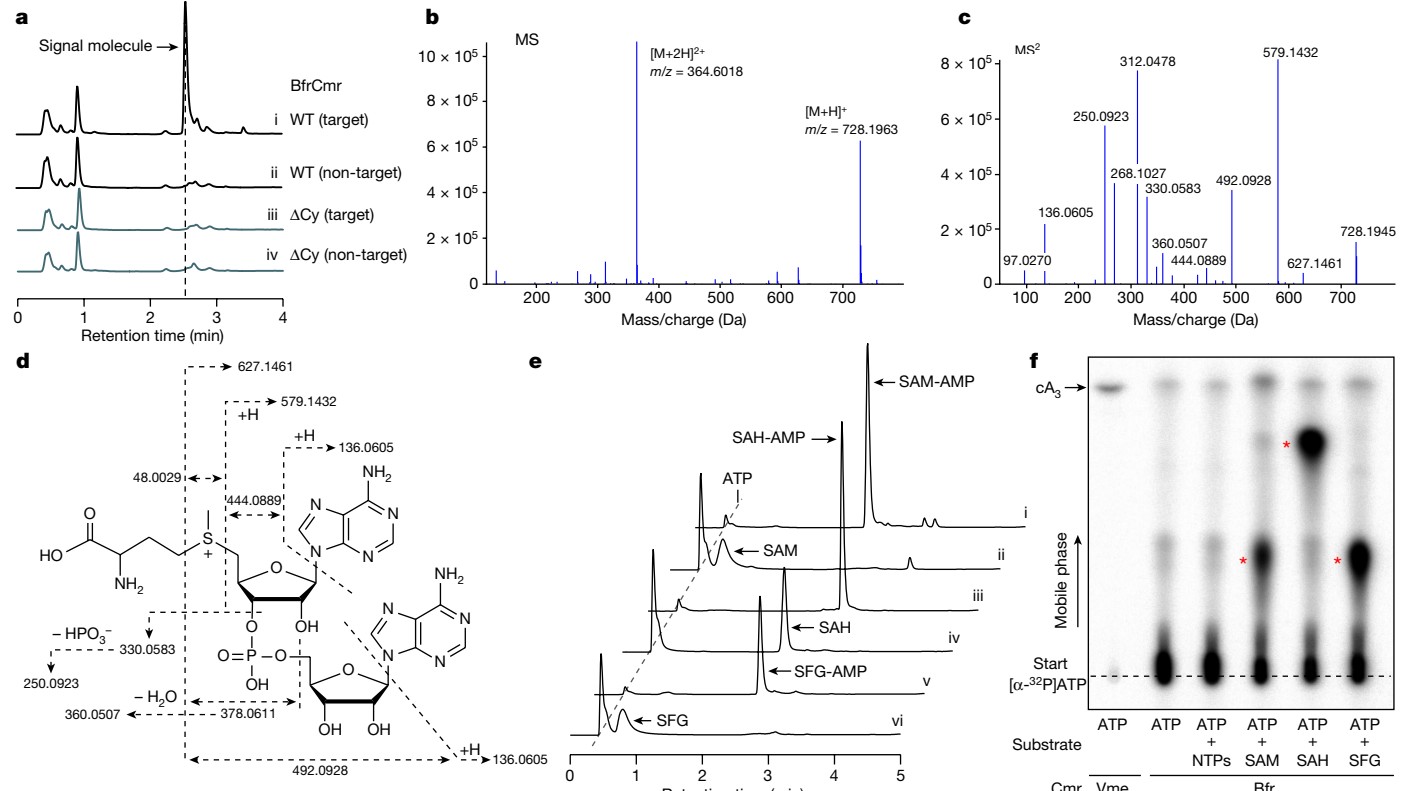

**Fig. 2 | Identification of SAM-AMP from cells containing the activated Cmr complex. a**, HPLC analysis of *E. coli* extracts expressing the wild-type (WT) or mutant (ΔCy) *B. fragilis* Cmr system with target or non-target crRNA. The putative signal molecule was only observed for the activated system (trace i). **b**, Characterization of the extracted signal molecule by liquid chromatography–mass spectrometry (LC–MS) in positive mode. [M + H]⁺ and [M + 2H]²⁺ represent two different ionization forms. **c**, MS/MS analysis of the signal molecule with *m/z* 728.1963. **d**, The proposed structure of the signalling molecule, whose fragmentation pattern is shown by dashed arrows. The MS/MS data cannot distinguish between 2′–5′ and 3′–5′ phosphodiester bonds.

The 3′–5′ phosphodiester bonds are more likely and is shown here, but a 2′–5′ bond cannot be completely ruled out. **e**, HPLC analysis of compounds synthesized by the purified wild-type *B. fragilis* Cmr complex in vitro. Cmr synthesizes the signal molecule SAM-AMP from ATP and SAM (trace i). Cmr also accepts SAH and sinefungin (SFG) as substrates (traces iii and v, respectively). Traces ii, iv and vi are control reactions. **f**, TLC analysis of in vitro reaction products. SAM, SAH and sinefungin plus ATP yielded radioactive products (red stars) but ATP alone did not. cA₃ generated by wild-type *V. metoecus* Cmr complex[15] is shown for comparison. Uncropped HPLC and TLC data are presented in Supplementary Data Fig. 2.

activity (Fig. 3c,d). Moreover, the double mutant displayed an enhanced pppApA synthase activity when compared with the wild-type enzyme, suggesting a partial reversion of the acceptor binding site to favour ATP over SAM (Extended Data Fig. 7). A deeper understanding of the reaction mechanism and substrate specificity of the SAM-AMP synthases will require structural data in the presence of ligands, and could also involve discrimination by the Cas5 subunit, which is in the vicinity of the ATP2 ligand.

## SAM-AMP signalling and turnover

To test the suggestion that SAM-AMP is the activator of the CorA effector, we expressed *B. fragilis* CorA in *E. coli* and purified the protein to near homogeneity in the presence of detergent (Extended Data Fig. 8a). CorA was incubated with radiolabelled SAM-AMP, SAH-AMP or cA₃ and then analysed by native gel electrophoresis. A clear shifted species, close to the wells of the gel, was observed to accumulate as the CorA protein was incubated at increasing concentrations with SAM-AMP or SAH-AMP (Fig. 4a). By contrast, cA₃ was not shifted. These data support a model where CorA binds the SAM-AMP second messenger to provide immunity. To investigate this in more detail, we generated a model of the pentameric *B. fragilis* CorA structure (Fig. 4b and Extended Data Fig. 8) and mapped the positions of conserved residues in the CorA-1 clade identified from a multiple sequence alignment (Supplementary

Data Fig. 10). A cluster of conserved residues at the interdomain interface hinted at a putative SAM-AMP-binding site. To test this, we created two site-directed variants of CorA by mutating two pairs of conserved residues (R152–R153 and D219–D220) in this cluster to alanine. The variant proteins were expressed similarly to the wild-type CorA, but no longer provided immunity in the plasmid challenge assay (Extended Data Fig. 8). Although these observations are consistent with a role in SAM-AMP binding, we cannot rule out the possibility that these mutations alter the quaternary structure of the protein. Although the mechanism of the CorA effector has not yet been determined, it most probably functions as a SAM-AMP-activated membrane channel, analogous to the Csx28 protein associated with Cas13[35] and to a number of other predicted membrane proteins associated with type III CRISPR systems[16]

As described previously, most type III systems with a CorA effector also encode a PDE of the NrN or DEDD family[16]. We therefore incubated the *B. fragilis* NrN protein with SAM-AMP and observed that it specifically degrades SAM-AMP (Fig. 4c), but not the linear dinucleotide pApA, or cOA molecules cA₂₋₆ (Extended Data Fig. 9a). One possibility is that specialized NrN and DEDD-family PDEs represent a type of 'off switch' to reset the system, analogous to the ring nucleases that degrade cOA molecules in canonical type III CRISPR systems[10]. In the *Clostridia*, the NrN protein is replaced with a predicted SAM lyase (Fig. 1b), suggesting an alternative means to degrade the SAM-AMP signalling molecule.

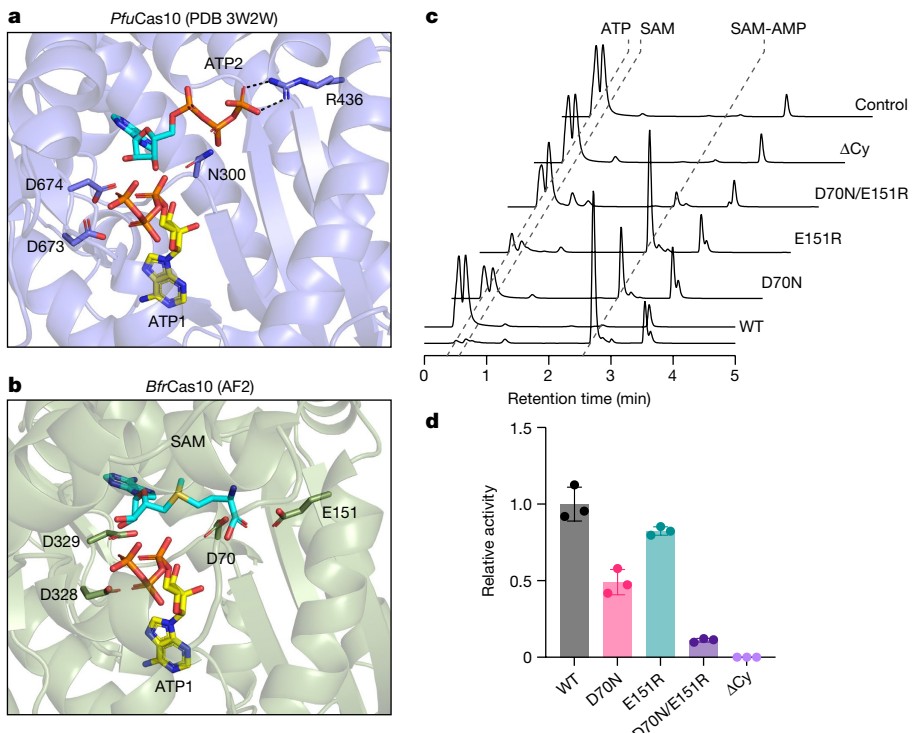

**Fig. 3 | Changes in the active site of Cas10 during synthesis of SAM-AMP.**
**a**, The crystal structure of the *P. furiosus* (*Pfu*) Cas10 subunit with 2 bound ATP molecules[33]. Side chains for the two metal binding aspartate residues of the DD motif, together with residues N300 and R436 that interact with ATP2, are shown. **b**, Equivalent view of the AF2 model of the *B. fragilis* (*Bfr*) Cas10 structure with ATP1 from the *P. furiosus* Cas10 structure and ATP2 replaced by SAM. The precise conformation and position of SAM is unknown. The conserved acidic residues D70, E151, D328 and D329 are shown. **c**, In vitro SAM-AMP synthase activity of wild-type and variant *B. fragilis* Cmr, analysed by HPLC following incubation of 2 μM Cmr with 0.5 mM ATP and SAM for 30 min. Raw HPLC data are presented in Supplementary Data Fig. 3. **d**, Relative SAM-AMP synthase activity of Cmr variants. Three independent experiments; data are mean ± s.d., calculated using GraphPad Prism 9.

We tested this by cloning and expressing the SAM lyase from *C. botulinum* and measuring its ability to degrade SAM-AMP, observing efficient degradation of the molecule to 5′-methylthioadenosine (MTA) (Fig. 4c,d). The other product of a lyase reaction, L-homoserine lactone, is not detectable by UV. The *C. botulinum* lyase degrades SAM-AMP more efficiently than SAM (Extended Data Fig. 9b), consistent with a specialized role in defence.

## Discussion

The polymerase active site of Cas10, the catalytic subunit of type III CRISPR systems, which consists of two DNA polymerase family B palm domains, is known to synthesize a range of cOA second messengers for antiviral defence. Here we have shown that some type III CRISPR systems signal via synthesis of SAM-AMP, a previously unknown molecule created by the adenylation of SAM (Fig. 5), which thus represents a novel nucleotide-based second messenger. In bacteria, the most recently discovered anti-phage signalling molecules include the cUMP and cCMP of the PYCSAR system[36], a wide range of cyclic di- and tri-nucleotides of the CBASS system[37] and the cOAs typically made by type III CRISPR systems[6,7]. Given the structural similarity between ATP and SAM, it is perhaps not surprising that SAM can substitute for ATP as an acceptor for a new 5′–3′ phosphodiester bond in the active site of nucleotide cyclases, following limited sequence divergence. Clearly this reaction reaches a natural end point as there is no possibility of cyclization or further polymerization. Judging by the distribution of CorA effectors, SAM-AMP signalling has a patchy but wide distribution in members of the *bacteroidetes*, *firmicutes*, *δ-proteobacteria*, *ε-proteobacteria* and *euryarchaea*. This is consistent with the high levels of defence system gain (by lateral gene transfer) and loss observed

generally, and may be a reflection of the pressures exerted by viruses, driving diversity.

CRISPR-associated CorA proteins are predicted to have a N-terminal soluble domain fused to a C-terminal transmembrane helical domain related to the CorA family of divalent cation transporters[16]. We postulate that binding of SAM-AMP to the cytoplasmic domain results in an opening of the transmembrane pore to effect immunity, but alternative mechanisms of membrane disruption have been observed for bacterial immune effectors[38], so this is a priority for future studies. The CorA effectors seem to be obligately associated with degradative enzymes such as NrN in *B. fragilis*, sometimes even being fused[16]. The observation that SAM-AMP is easily purified from extracts of *E. coli* expressing the activated *B. fragilis* Cmr system suggests that SAM-AMP is not a substrate for the generalist ribonucleases present in bacteria, necessitating the addition of a specialized PDE such as NrN. In these respects, NrN is reminiscent of the ring nucleases (Crn1–3 and Csx3) that are frequently found associated with cOA generating CRISPR systems[10]. This suggests that it is beneficial to the cell to deplete the SAM-AMP signalling molecule, perhaps to avoid unnecessary cell death when phage infection has been cleared. In this regard, it is telling that the NrN PDE is sometimes replaced by a SAM-AMP lyase—an enzyme that degrades SAM-AMP using an entirely different mechanism[23], yielding different products. SAM lyases are typically phage related genes and are thought to function by neutralizing DNA methylases in host restriction-modification systems[23,24]. In the context of CorA-family CRISPR systems, SAM-AMP lyases encoded by mobile genetic elements may also function as anti-CRISPRs, similarly to viral ring nucleases[39].

Some important open questions remain. It is difficult to explain the toxicity of CorA when no SAM-AMP is synthesized and NrN is absent,

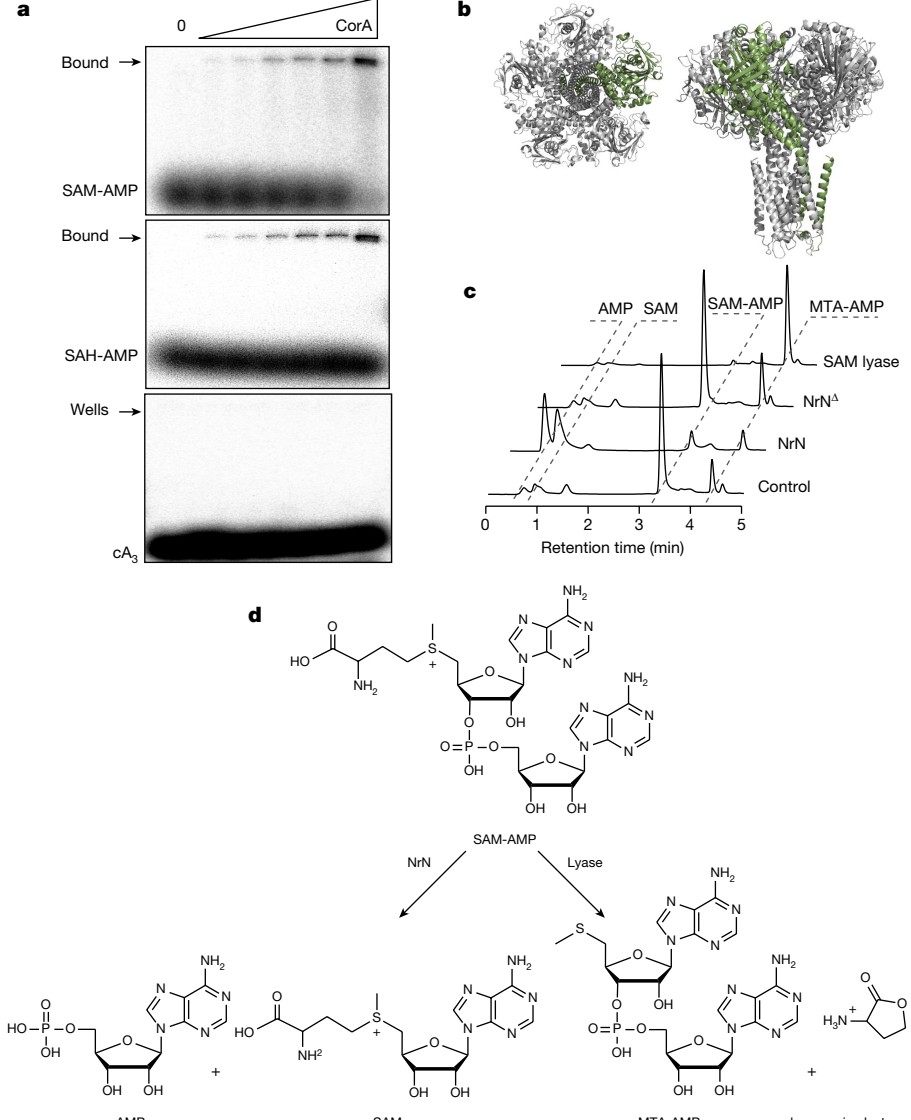

**Fig. 4 | Binding and degradation of SAM-AMP by ancillary CRISPR proteins.**
**a**, CorA binds SAM-AMP and SAH-AMP, but not cA$_3$ (1 μM $^{32}$P-labelled ligand incubated with 0, 0.0625, 0.125, 0.35, 0.75, 1.5 and 3.3 μM CorA), illustrated by acrylamide gel electrophoresis and phosphorimaging. Uncropped gels are shown in Supplementary Data Fig. 4. **b**, Two views of the predicted structure of the pentameric CorA channel, with one subunit coloured green. **c**, NrN specifically degrades SAM-AMP to SAM and AMP. HPLC analysis of samples in which purified SAM-AMP was incubated with NrN and NrN$^Δ$, an inactive variant with D85A/H86A/H87A mutations. *C. botulinum* lyase degrades SAM-AMP to generate MTA and L-homoserine lactone (not UV visible). Small amounts of MTA are present in the SAM-AMP sample purified from *E. coli*. Uncropped HPLC traces are available in Supplementary Data Fig. 4. **d**, Schematic representation of the reactions catalysed by NrN and SAM-AMP lyase.

as well as the observation that both CorA and NrN are required for immunity. These data suggest a close functional link between CorA and NrN, although we have detected no physical interaction between the two proteins in vitro. Rather than functioning in a manner analogous to ring nucleases, an alternative hypothesis is that NrN (or SAM-AMP lyase) is required to prevent de-sensitization of the CorA channel—a phenomenon observed for other pentameric ligand-gated ion channels when activator concentrations remain high[40]. Answers to these questions will probably require further analysis of the system in a cognate host at native expression levels, coupled with structure–function studies of the CorA channel.

Given the wide range of SAM and ATP analogues available, the discovery of an enzymatic route to synthesis of SAM-AMP opens the way to the generation of a new family of bioactive molecules. For example, there is considerable interest in the development of specific inhibitors of methyltransferases, a large family of enzymes (more than 300

methyltransferases are encoded in the human genome) involved in many key cellular reactions[41]. Depending on the specificity of the Cas10 enzyme, a range of SAM and ATP analogues could be provided as building blocks to make a diverse family of SAM-AMP analogues with altered properties. As we have seen, replacement of the methyl group on the sulfur atom of SAM with a proton (in SAH) or an amino group (sinefungin) still supports catalysis by *B. fragilis* Cas10. Many other modification sites are available on the parental molecules.

In conclusion, we report the discovery of SAM-AMP, which is synthesized from two of the most abundant molecules in the cell and functions as a second messenger of viral infection. This broadens the repertoire of type III CRISPR systems and may have implications for immune signalling more generally, as family B polymerases are a widespread and diverse superfamily found in all branches of life. The recent expansion of our knowledge of signalling molecules reflects the fact that Nature tends to use and repurpose such molecules in diverse cellular processes.

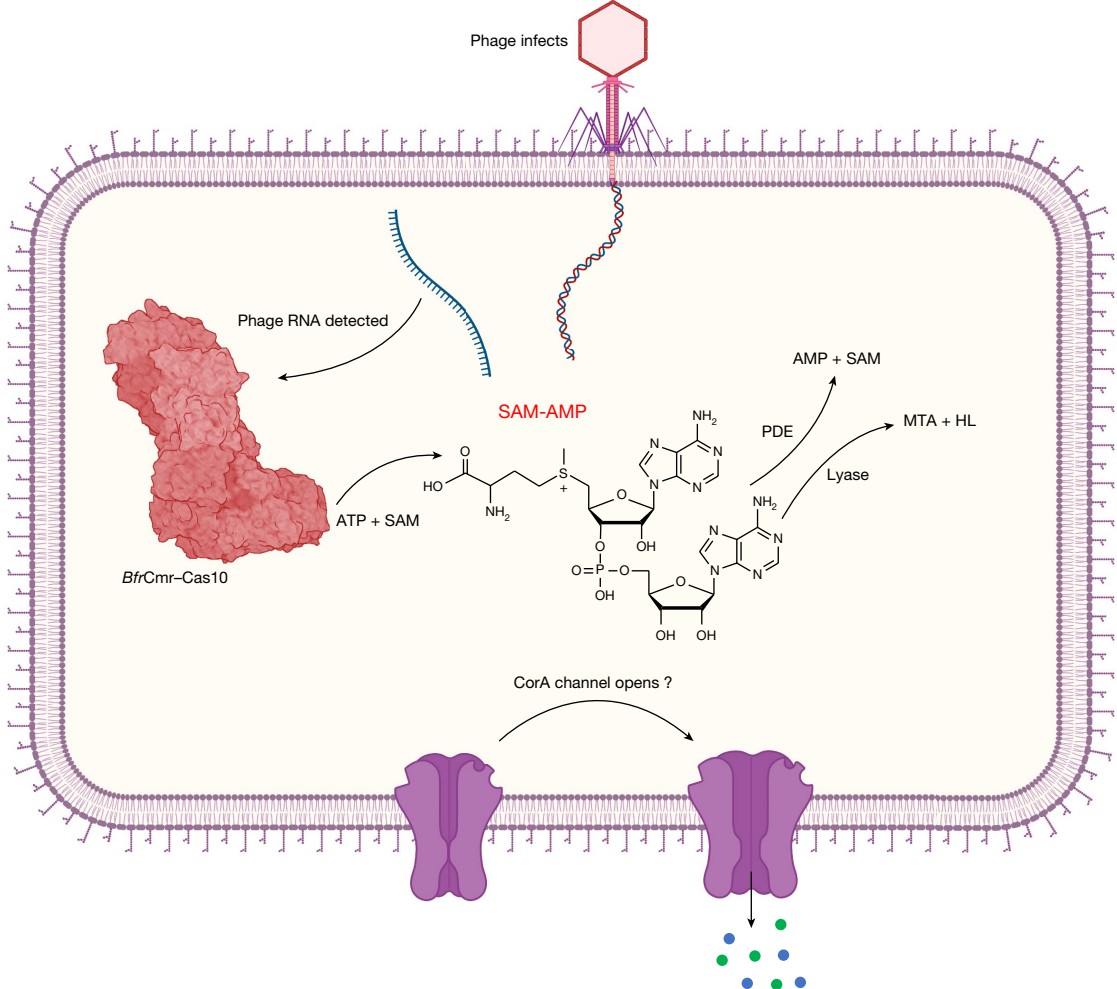

**Fig. 5 | Model of the SAM-AMP immune signalling pathway.** Transcription of the infecting phage genome activates the *B. fragilis* Cmr complex, resulting in synthesis of the SAM-AMP second messenger. SAM-AMP binds to the CorA membrane protein, resulting in the opening of a pore that disrupts the host membrane to combat infection. SAM-AMP is degraded by specialized PDE enzymes that hydrolyse the phosphodiester bond, generating AMP and SAM or lyases that target the methionine moiety, generating MTA and homoserine lactone (HL). These enzymes are likely to deactivate the signalling molecule to reset the system once phage have been eliminated. Created with BioRender.com.

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

## Methods

### Cloning

Supplementary Table 1 shows the synthetic gene, DNA and RNA oligonucleotide sequences used in this study.

The synthetic genes encoding *B.fragilis* Cas6, CorA, NrN and *C. botulinum* SAM lyase purchased as g-blocks (IDT) were codon-optimized for expression in *E. coli* C43 (DE3) via the vector pEhisV5TEV, which encodes eight histidines, a V5 epitope tag and the cleavage site of Tobacco Etch Virus (TEV) protease[43].

The pACE-based Cmr expression plasmid pBfrCmr1-6 was designed to contain six codon-optimized genes *cmr1–6* (Twist Biosciences), with a polyhistidine tag on the N-terminus of Cmr3. We then divided pBfrCmr1-6 into five overlapping segments with similar length (designated as BfrCmr a, b, c, d and e). These segments were amplified by PCR using primers BfrCmrSG-f and BfrCmrSG-r (Supplementary Table 1) and then assembled into pBfrCmr1-6 using NEBuilder HiFi DNA Assembly Master Kit. The obtained plasmid pBfrCmr1-6 was verified by restriction digest and sequencing.

For the construction of the crRNA over-expression vector, the codon-optimized *cas6* gene was synthesized as a g-Block (IDT) and inserted into the NdeI and XhoI restriction sites in MCS-2 of the vector pCDFDuet. The synthetic gene of CRISPR pre-array with two CRISPR repeats and two divergent BpiI sites between two repeats for a spacer sequence insertion was cloned into 5′ NcoI and 3′ SalI sites in MCS-1 of pCDFDuet containing *cas6*. Spacers targeting the tetracycline resistance gene *tetR* or *lacZ* were ligated into the BpiI sites of CRISPR pre-array to obtain the plasmid, designated as pCRISPR_Tet or pCRISPR_pUC. A CRISPR array, consisting of one spacer targeting the gene encoding late promoter activating protein (Lpa) of phage P1, flanked by two *B. fragilis* CRISPR repeat sequences, was assembled by annealing primers Bfr-rep-5p-T, Bfr-rep-5p-C, Bfr-rep-3p-T, Bfr-rep-3p-C, Bfr-sp-phageIPA-T and Bfr-sp-phageIPA-C (Supplementary Table 1). The array was then ligated into MCS-1 of pCDFDuet containing *cas6* in MCS−2 to give pCRISPR_Lpa.

To construct pRAT-Duet-derived plasmids, for single effector expression, synthetic genes encoding CorA, NrN or their variants were inserted into NcoI and EcoRI sites in MCS-1 under control of the pBAD promoter. The *nrn* gene was cloned into the NdeI and XhoI site in MCS-2 of pRAT-Duet containing *corA* in MCS-1 for effector co-expression.

For mutagenesis, all variants were generated by PCR using Phusion enzyme (Thermo Scientific) with wild-type constructs as the templates in presence of two overlapping primers containing the target mutations. The correct variants were confirmed by sequencing.

### Protein production and purification

*B. fragilis* NrN, Cas6 and *C. botulinum* SAM lyase were expressed and purified using standard procedures described previously, with removal of the N-terminal his-tags by TEV protease[43]. The Cmr complex with crRNA expressed from cells co-transformed with pACE-BfrCmr and pCRISPR_Lpa was purified using the same procedures. The purification of membrane protein BfrCorA was followed the similar procedures. In brief, after induction of CorA expression with 0.2 mM IPTG, the cells were harvested and lysed in the lysis buffer (50 mM HEPES pH 7.5, 250 mM NaCl, 5% glycerol, 10 mM imidazole). The membrane fraction was isolated by ultra-centrifuge at 40,000 rpm at 4 °C for 2 h and then resuspended in the lysis buffer supplement with 1% *n*-dodecyl-β-D-maltopyranoside (Glycon Biochemical) and incubated at 4 °C for 1 h. The solubilised CorA was purified by nickel affinity chromatography and size-exclusion chromatography, where buffers contained 0.03% *n*-dodecyl-β-D-maltopyranoside. Subsequently, a series of variants were constructed and purified following the same procedure as the wild type.

### Plasmid challenge assay

pBfrCmr1–6 and pCRISPR_Tet (or pCRISPR_pUC) were co-transformed into *E. coli* BL21 Star cells. Single colonies were picked for competent cell preparation into L-Broth (100 μg ml⁻¹ ampicillin and 50 μg ml⁻¹ spectinomycin) and cultivated at 37 °C overnight. Overnight culture was diluted 50-fold into 20 ml selective LB medium and grown at 37 °C with shaking at 220 rpm until the $OD_{600}$ reached 0.4–0.5. Cell pellets were collected and then resuspended in an equal volume of pre-chilled competent cells solution (60 mM $CaCl_2$, 25 mM MES, pH 5.8, 5 mM $MgCl_2$, 5 mM $MnCl_2$). Cells were incubated on ice for 1 h, centrifuged and the collected pellet was resuspended in 0.1 volumes of the same buffer containing 10% glycerol. Aliquots (100 μl) were flash frozen by liquid nitrogen and then stored at −80 °C. 100 ng pRAT or pRAT derived plasmids carrying the target gene were added to the competent cells, incubated on ice for 30 min and transformed by heat shock. Following addition of 0.5 ml LB medium, the transformation mixture was incubated at 37 °C for 2.5 h. 3 μl of a 10-fold serial dilution was applied in duplicate to LB agar plates (supplemented with 100 μg ml⁻¹ ampicillin and 50 μg ml⁻¹ spectinomycin) for the uninduced condition. The transformants were selected on LB agar containing 100 μg ml⁻¹ ampicillin, 50 μg ml⁻¹ spectinomycin and 12.5 μg ml⁻¹ tetracycline. Addition of 0.2% (w/v) lactose and 0.2% (w/v) L-arabinose was used for full induction. Plates were incubated at 37 °C overnight. The experiment was performed as two independent experiments with two biological replicates and at least two technical replicates.

### Synthesis of SAM-AMP and its analogues

For in vitro synthesis, 2 μM wild-type Cmr was incubated with ATP and SAM, SAH or sinefungin (0.1 mM each for radiolabelled products or 0.5 mM each for HPLC analysis) in reaction buffer (20 mM Tris-HCl, pH 7.5, 10 mM NaCl, 1% glycerol and 5 mM $MnCl_2$). The reaction was initiated by adding 5 μM target RNA–Lpa (using non-target RNA-pUC as negative control) and carried out at 37 °C for 1 h or the times indicated. 4 nM $\alpha^{32}$P-ATP as a tracer was added in each reaction to generate radiolabelled products, if needed for TLC analysis or electrophoretic mobility shift assay (EMSA).

For in vivo production, a single colony of *E. coli* BL21 Star cells transformed with the plasmids pBfrCmr1-6, pCRISPR_Tet (or pCRISPR_pUC) and pRAT-Duet was inoculated into 10 ml of L-broth with antibiotics (50 μg ml⁻¹ ampicillin, 50 μg ml⁻¹ spectinomycin and 12.5 μg ml⁻¹ tetracycline) and grown overnight at 37 °C with shaking at 180 rpm. 20-fold overnight culture was diluted into 20 ml fresh L-broth with same antibiotics and then incubated at 37 °C. The culture was fully induced with 0.2% (w/v) D-lactose and 0.2% (w/v) L-arabinose after reaching an $OD_{600}$ between 0.4 and 0.6. After overnight induction at 25 °C, the cell culture was mixed with 4 volumes of cold PBS and then centrifuged at 4,000 rpm for 10 min at 4 °C. Cell pellet was resuspended into 2 ml cold extraction solvent (acetonitrile:methanol:water, 2:2:1 by volume), vortexed for 30 s and stored at −20 °C until needed. The supernatant was obtained by centrifuging at 13,200 rpm for 10 min at 4 °C, followed by evaporation until samples were completely dry, then resuspended in water and analysed by HPLC or LC–MS. HPLC data were collected and analysed using Chromeleon 6.8 Chromatography Data System software (Thermo Fisher).

### Treatment with nuclease P1

For nuclease treatment, 100 μM of compound was incubated with 0.02 units P1 nuclease (New England Biolabs) in P1 reaction buffer (50 mM sodium acetate pH 5.5) at 37 °C for 1 h. Each reaction solution was deproteinised with a spin filter (Pall Nanosep®, MWCO 3 kDa) followed by HPLC analysis.

### Liquid chromatography and mass spectrometry

Enzymatic reactions were analysed on an UltiMate 3000 UHPLC system (Thermo Fisher scientific) with absorbance monitoring at 260 nm.

Samples were injected into a C18 column (Kinetex EVO 2.1 × 50 mm, particle size 2.6 µm) at 40 °C. Gradient elution was performed with solvent A (10 mM ammonium bicarbonate) and solvent B (acetonitrile plus 0.1% TFA) at a flow rate of 0.3 ml min$^{-1}$ as follows: 0–0.5 min, 0% B; 0.5–3.5 min, 20% B; 3.5–5 min, 50% B; 5–7 min, 100% B.

LC–MS and LC–MS/MS analysis were carried out on a Eksigent 400 LC coupled to Sciex 6600 QTof mass spectrometer, in trap elute configuration at micro flow rates. Samples were loaded onto a YMC Triart C18 trap cartridge 0.5 ×5.0 mm in 99.95% water, 0.05% TFA at 10 µl min$^{-1}$. After 3 min washing the salts to waste, the trap was switched in-line with the analytical column: a YMC Triart 150 ×0.075 mm. Gradient elution was performed with solvent A (99.9% water, 0.1% formic acid) and solvent B (20% water 80% acetonitrile 0.1% formic acid) at a flow rate of 5 µl min$^{-1}$ as follows: 0 min 3% B; 0–6 min 95% B; 6–8 min 95% B; 8–9 min 3% B; 9–13 min re-equilibrate 3% B. The flow from the column sprayed directly into the ESI turbospray orifice of the mass spectrometer. Data were collected in positive ionization mode from 120–1,000 $m/z$. Ions of interest were selected for CID fragmentation at collision voltages of 25-45 V, and the fragmentation spectra collected from 50–1,000 $m/z$. The mass spectrometer was externally calibrated prior to analysis with Sciex tuning solution 4457953.

## Thin-layer chromatography
Radio-labelled SAM-AMP and its analogues were separated by TLC. 1 µl radiolabelled products were analysed on 20 × 20 cm Silica gel TLC aluminium plates (Sigma-Aldrich) with 0.5 cm of TLC buffer (0.2 M ammonium bicarbonate, 70% ethanol, and 30% water pH 9.3) at 35 °C. The TLC plate was left in the TLC chamber until the solvent front was approximately 5 cm from the top of the TLC plate and finally phosphorimaged using a Typhoon FLA 7000 imager (GE Healthcare).

## Electrophoretic mobility shift assay
In total, 40 nM $^{32}$P-radiolabelled SAM-AMP, SAH-AMP or cA$_3$ was incubated with varying amounts of purified *B. fragilis* CorA (0.1, 0.2, 0.4, 0.8, 1.6 and 3.3 µM) in binding buffer (12.5 mM Tris-HCl, pH 8.0, 10% (v/v) glycerol, 0.5 mM EDTA) at 25 °C for 15 min. Reactions were mixed with ficoll loading buffer and then analysed on the native polyacrylamide gel (8% (w/v) 19:1 acrylamide:bis-acrylamide). Electrophoresis was carried out at 200 V for 2 h at room temperature using 1× TBE buffer as the running buffer, followed by phosphorimaging (Typhoon FLA 7000 imager (GE Healthcare), photomultiplier tube setting = 700–900).

## SAM-AMP degradation assay
SAM-AMP degradation assay was carried out by incubating 1 µM wild-type *B. fragilis* NrN or *C. botulinum* SAM lyase and their inactive variants with 100 µM purified SAM-AMP, its analogues or standards like SAM, pppApA, pApA or cOA mixture (cA$_2$, cA$_3$, cA$_4$ and cA$_6$), respectively in buffer (20 mM Tris-HCl pH 7.5, 20 mM NaCl, 1% glycerol and 0.5 mM MnCl$_2$) at 37 °C for 1 h or at indicated time points for time course assay. The reaction was stopped by mixing with 2 volumes pre-chilled methanol and vortexing for 30 s, before centrifuging at 13,000 rpm at 4 °C for 20 min to remove denatured protein. The supernatant was dried and then resuspended in the water, followed by HPLC or LC–MS/MS analysis.

## Western blotting for detection of *B. fragilis* CorA
For detection of *B. fragilis* CorA wild-type and variants expression, proteins were expressed from the pEV5hisTEV plasmid in *E. coli* C43 (DE3). The cells were grown in 5 ml LB containing 50 µg ml$^{-1}$ kanamycin at 37 °C until OD$_{600}$ reached 0.6–0.8, followed by 0.2 mM IPTG induction. After growth at 16 °C for 16 h, the pellets were collected by centrifugation at 4,000 rpm for 10 min and then resuspended in 1 ml lysis buffer (50 mM HEPES, pH 7.5, 250 mM NaCl, 5% glycerol, 10 mM imidazole). Ten microlitres of 20-times-diluted lysate was loaded onto a precast NuPAGE Bis-Tris Gel (Thermo Fisher Scientific) for separation and transferred into a nitrocellulose membrane using iBlot 2 Dry Blotting System (Invitrogen). Membranes were blocked for at least 1 h with shaking in 0.03% milk in TBST (20 mM Tris, 150 mM NaCl, pH 7.6, 0.1% Tween-20) and then incubated with mouse anti-V5 antibody (Invitrogen, cat. no. R960-25, clone SV5-Pk1) diluted 1:10,000 in 0.03% milk in TBST buffer at 4 °C with overnight shaking. Membranes were washed three times with 0.03% milk in TBST and then incubated with anti-mouse IgG antibody (LI-COR Biosciences) at 1:20,000 dilution with 0.03% milk in TBST for 1 h at room temperature with shaking. The membranes were washed again with 0.03% milk in TBST twice and TBST once, before imaging on an Odyssey imager system (LI-COR Biosciences).

## Nuclease assay
Nuclease activity of Cas6 was assayed by incubating 1.2 µM *B. fragilis* Cas6 with 300 nM 5′ end FAM-labelled *B. fragilis* CRISPR repeat RNA (Table S1, purchased from Integrated DNA Technologies (IDT)) in reaction buffer (20 mM Tris-HCl pH 7.5, 50 mM NaCl and 1 mM EDTA), at 37 °C for 5 min. The reaction was stopped by heating at 95 °C for 5 min and then analysed by 20% acrylamide, 7 M urea, 1× TBE denaturing gel, which was run at 30 W, 45 °C for 2 h. An alkaline hydrolysis ladder was generated by incubating RNA in 5 mM NaHCO$_3$, pH 9.5 at 95 °C for 5 min. The gel was finally imaged using a Typhoon FLA 7000 imager (GE Healthcare) at a wavelength of 532 nm (PMT = 600–700).

An internally radiolabelled transcript RNA containing two CRISPR repeats and one guide sequence (Supplementary Table 1) was incubated with 2 µM Cas6 in the same condition mentioned above and the reaction products were checked on the 20% acrylamide denaturing gel at different time points. The transcript was generated by following the instruction of MEGAscript®Kit (Invitrogen). The template used in transcription was obtained by PCR of plasmid pCRISPR_Lpa using primer Duet-up and Duet-Down (IDT; Supplementary Table 1). PCR product (120 ng) mixed with ATP, GTP, UTP, CTP solution and 133 nM α$^{32}$P-ATP as a tracer was incubated at 37 °C for 4 h in the 1× reaction buffer with T7 enzyme mix. Transcript was then purified by phenol: chloroform extraction and isopropanol precipitation.

## Target RNA cleavage assay
The 5′ end labelled RNA–Lpa was generated as described previously[15]. RNA cleavage assays using 1 µM wild-type Cmr (or variant with Cmr4 D27A) and 5′-end-labelled RNA–Lpa substrates were conducted in reaction buffer (20 mM Tris-HCl, pH 7.5, 10 mM NaCl, 1% glycerol and 5 mM MnCl$_2$, 0.1 U µl$^{-1}$ SUPERase•In (Thermo Scientific)) at 37 °C. The reaction was stopped at indicated time points by adding EDTA (pH 8.0) and extracted with phenol–chloroform to remove protein. After adding equal volume 100% formamide, the samples were loaded onto 20% denaturing polyacrylamide sequencing gel. The gel electrophoresis was carried out at 90 W for 3–4 h. Visualization was achieved by phosphorimaging as above. A 5′-end-labelled RNA–Lpa was subjected to alkaline hydrolysis generating a single nucleotide resolution ladder for RNA size determination.

## Bioinformatic analyses
To investigate the phylogenetic diversity of Cas10 proteins across type III CRISPR–Cas loci and their associations with CorA, 50,924 bacterial and 992 archaeal genomes from NCBI, marked as 'complete', were downloaded on 14 March 2023. The genomes were filtered for the presence of Cas10 by performing a hidden Markov model (HMM) search using Hmmer 3.3.2[44] and previously published Cas10 HMM profiles[42]. Hits with an E-value less than 10$^{-20}$ and minimum protein length above 500 AA were considered. The Cas10s from the resulting 2,209 genomes were clustered with CD-hit (similarity cutoff 0.80

and word size 5) resulting in 613 representative Cas10 sequences. All CRISPR–Cas loci in the associated 613 genomes were annotated with CRISPRCasTyper 1.8.0[42] and type III loci (or hybrid loci with type III as a component) that had an interference completeness percentage of over 30% were picked for further analysis. This threshold was chosen to exclude any solo Cas10 proteins, but also to include small CRISPR–Cas loci that may include effector proteins such as CorA. From this point forward all CRISPR–Cas loci were treated as independent of their host, i.e multiple type III loci from a single host were permitted, resulting in 745 loci. Proteins within these loci (and 2 kb upstream and downstream of the CCTyper-reported locus boundaries) were searched for CorA using Hmmer 3.3.2[44] with HMM profiles derived from CCTyper[42] using an E-value cutoff of $10^{-20}$. To create the Cas10 phylogenetic tree, the Cas10 proteins from each locus were aligned using Muscle 5.1[45] with the Super5 algorithm intended for large datasets. The Cas10 phylogenetic tree was constructed from the alignment using FastTree 2.1.11[46] with the WAG + CA model and Gamma20-based likelihood. The tree was visualized in R 4.1.1 and RStudio 2021.9.0.351 (http://www.rstudio.com/) using ggtree[47] and ggplot2[48]. These steps were wrapped in a Snakemake 7.22.0[49] pipeline and an R script (available at Github: https://github.com/vihoikka/Cas10_prober).

## Structural modelling of *B. fragilis* Cas10 and *B. fragilis* CorA

The structure of the *B. fragilis* Cas10–Cas5 heterodimer and *B. fragilis* CorA monomer were predicted using AlphaFold2[50,51] (AF2) as implemented by the Colabfold server[52]. The structure of the C-terminal 172 amino acids of CorA, including the transmembrane domain, which is the region with detectable similarity to the CorA magnesium transporter, was modelled as a pentamer using Alphafold2. Five individual monomers were then superimposed on the pentameric transmembrane structure to yield a model of the full-length pentameric *B. fragilis* CorA structure. This approach avoided memory and processor constraints inherent in Colabfold for such a large modelling project. The local distance different test (LDDT) scores for all three models are shown in Extended Data Fig. 10.

## Statistics and reproducibility

The CorA binding assay in Fig. 4a was repeated as duplicates. Assays for nuclease activity of Cas6 (Extended Data Fig. 2) and target RNA cleavage activity of *B. fragilis* Cmr complex (Extended Data Fig. 3) were conducted as three independent experiments and representative examples are shown. The purification of CorA (Extended Data Fig. 8a) was repeated three times and the western blot of CorA (Extended Data Fig. 8e) was performed as two independent experiments (two biological replicates) with comparable results.

## Reporting summary

Further information on research design is available in the Nature Portfolio Reporting Summary linked to this article.

## Data availability

Mass spectrometry data are available on FigShare: https://doi.org/10.6084/m9.figshare.c.6646859.v1. Coordinates for experimentally determined protein structures (for example Protein Data Bank ID 3W2W) are available at https://www.rcsb.org/. The genome sequences used to make Fig. 1a were downloaded from NCBI (https://www.ncbi.nlm.nih.gov) on 14 March 2023.

## Code availability

Code used to generate the phylogenetic tree was wrapped in a Snakemake 7.22.0[49] pipeline and an R script that are available in Github: https://github.com/vihoikka/Cas10_prober.

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

**Acknowledgements** This work was supported by grants from the Biotechnology and Biological Sciences Research Council (Grant BB/T004789/1 to M.F.W.) and European Research Council (ref. 101018608 to M.F.W.). H.C. acknowledges the support of the China Scholarship Council (code 202008420207). V.H. is funded by the Finnish Cultural Foundation. The authors thank D. O'Hagan and R. White for helpful discussions. For the purpose of open access, the author has applied a CC BY public copyright Licence to any Accepted Author Manuscript version arising from this submission.

**Author contributions** H.C. planned, carried out and analysed the molecular biology, microbiology and biochemistry experiments and drafted the manuscript. V.H. carried out the bioinformatic analyses. S. Grüschow planned the protein expression and plasmid challenge experiments with H.C. S.Graham cloned genes and expressed proteins. S.S. carried out the mass spectrometry and analysed the data. M.F.W. conceptualized and oversaw the project, obtained funding and analysed data along with the other authors. All authors contributed to the drafting and revision of the manuscript.

**Competing interests** The authors declare no competing interests.

**Additional information**
**Correspondence and requests for materials** should be addressed to Malcolm F. White.

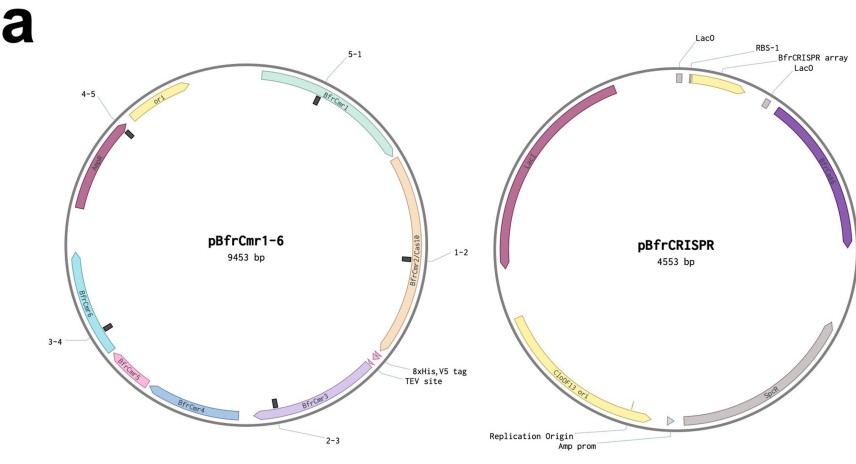

## b

### BfrCmr wild type

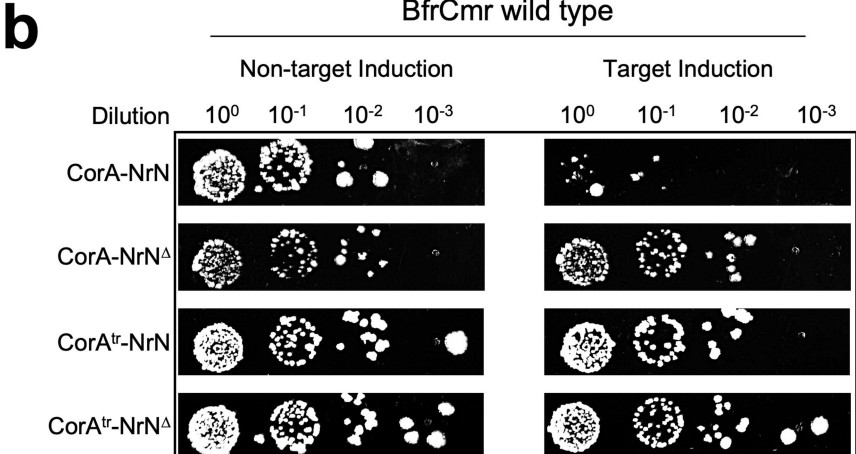

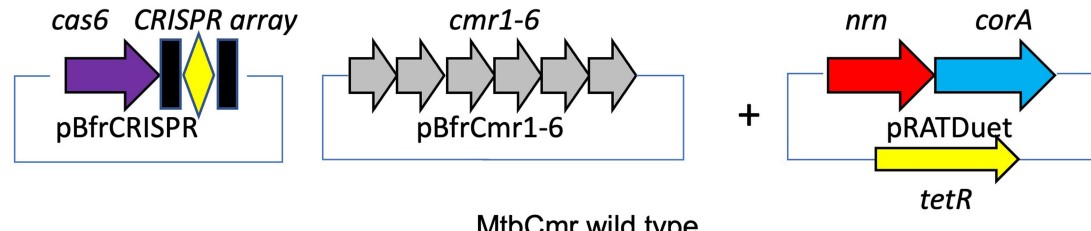

### MtbCmr wild type

## c

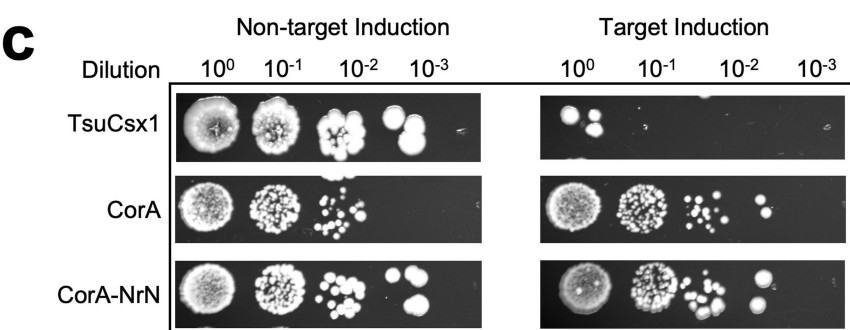

**Extended Data Fig. 1 | Plasmid challenge assay. a**, Plasmid pBfrCmr1-6 was constructed by Gibson assembly to express genes cmr1–cmr6 (junctions indicated by black squares); Plasmid pCRISPR was constructed to express the CRISPR array along with Cas6. Plasmid maps were generated by Benchling (https://benchling.com/). **b**, Different variants of ancillary effectors were tested in the *B. fragilis* Cmr wild type target (tetR) and non-target (pUC19) system. CorA$^{tr}$ is the truncated CorA (aa 1-428) and NrN$^{\Delta}$ is an inactive variant of NrN (D85A/H86A/H87A). The schematic shows cartoons of the three plasmids used in these experiments. **c**, The ancillary effectors CorA and NrN were assayed in the *Mycobacterium tuberculosis* (Mtb) Cmr wild type system. TsuCsx1 from *Thioalkalivibrio sulfidiphilus* (activated by cA$_4$) acted as a positive control. Uncropped images and gels are available in Supplementary Data Fig. 4.

## a

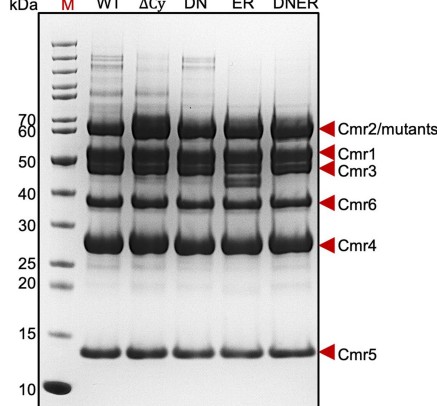

## b

## c

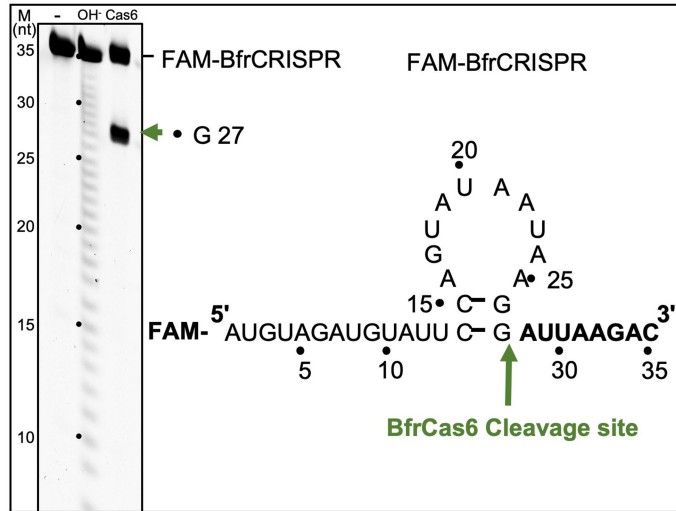

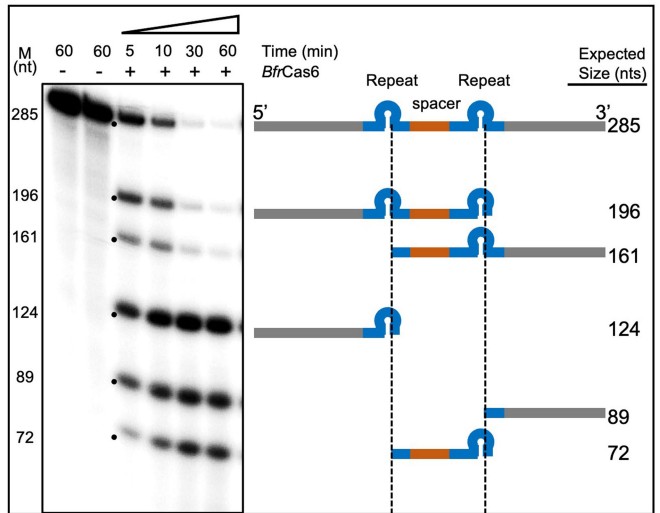

**Extended Data Fig. 2 | Expression of the *B. fragilis* Cmr complex and CRISPR repeat RNA processed by Cas6. a**, SDS-PAGE analysis of the purified wild type (WT) and variants of Cmr (1-6) complex which include Cmr2 D328A:D329A (Δcy), Cmr2 D70N (DN), Cmr2 E151R (ER) and the double mutant (D70N:E151R, DNER). **b**, The cleavage site of Cas6 within the CRISPR repeat RNA was mapped by incubating 5′ end FAM-labelled repeat RNA (300 nM) with Cas6 nuclease (1.2 µM). Alkaline hydrolysis (OH⁻) ladder was used to mark the size of 5′ RNA cleavage products (green arrow). Potential secondary structure of CRISPR repeat RNA

with cleavage site was indicated (green arrow). **c**, An internally radio-labelled transcript RNA containing two CRISPR repeats (blue) and one guide (targeting Phage P1) sequence (orange) was incubated with BfrCas6 (2 µM). Samples were collected at the indicated time points and analysed by denaturing gel. The expected sizes and compositions of cleavage products are indicated based on the specific cleavage site of Cas6 within each repeat. Uncropped gels are available in Supplementary Data Figs. 3–5.

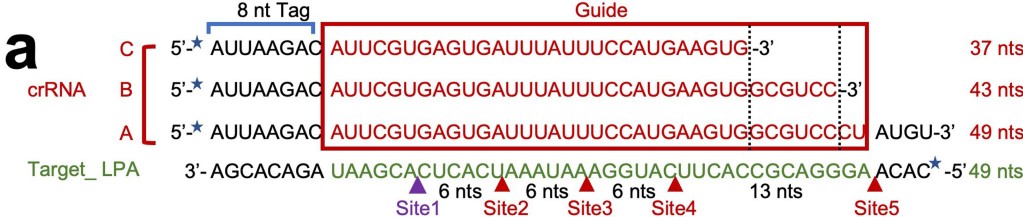

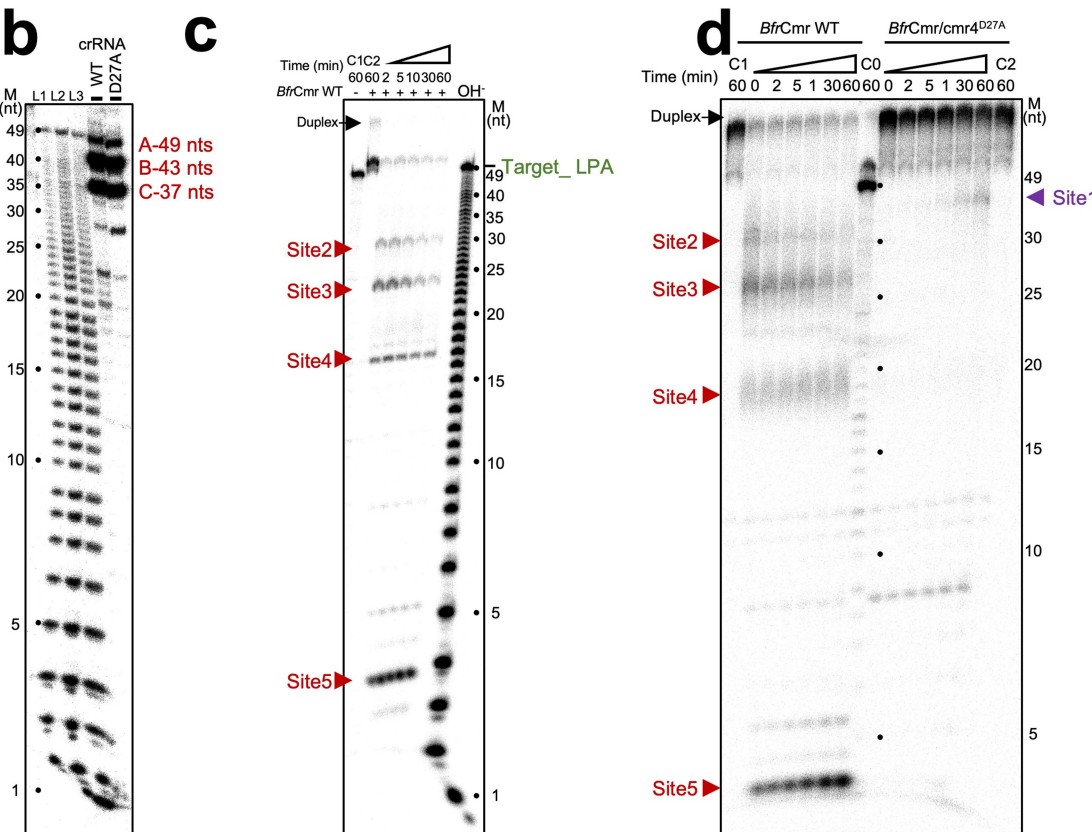

**Extended Data Fig. 3 | *B. fragilis* Cmr crRNA composition and target RNA degradation. a**, The sequence of crRNA species extracted from purified Cmr and the target RNA substrate used in the activity assay. The repeat-derived sequence (8 nt tag), spacer-derived sequence (guide) and the sequence complementary to guide RNA are coloured black, red and green, respectively. Five putative cleavage sites are indicated by arrows (Site1 is indicated by purple arrows, while sites 2 to 5 by red arrows). Extracted crRNAs and the target RNA substrate were 5′-labelled with 32 P (blue star). **b**, The size of extracted crRNAs from wild type and mutant Cmr (Cmr4 D27A) was mapped by comparing with

alkaline hydrolysis ladder of Target_LPA substrates (L1-3 with increased concentration of substrates). **c**, The indicated Target_LPA was incubated with (+) or without (−/C1) wild type Cmr in the presence of Mn²⁺ (no Mn²⁺ in buffer C2). The cleavage sites were mapped by comparing with alkaline hydrolysis (OH⁻) ladder and indicated by red arrows. **d**, Time course of cleavage on the 5′-radio-labelled Target_LPA by wild type or mutant Cmr (Cmr4 D27A). The buffer of C1 and C2 are in absence of Mn²⁺, while C0 is in absence of Cmr. Uncropped gels are available in Supplementary Data Fig. 5.

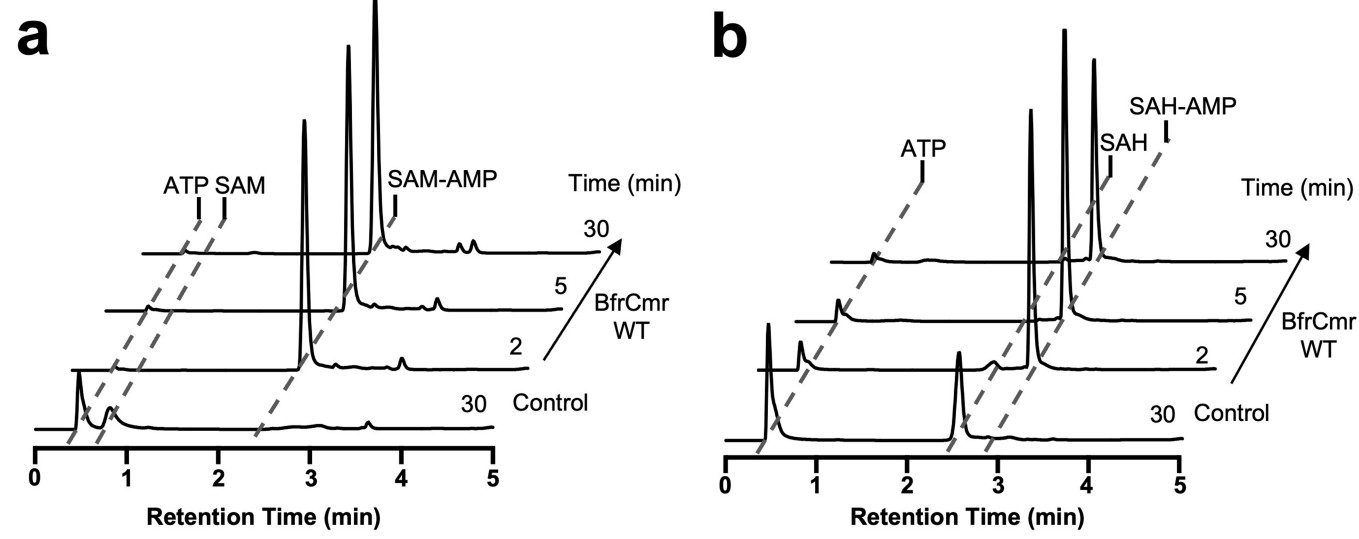

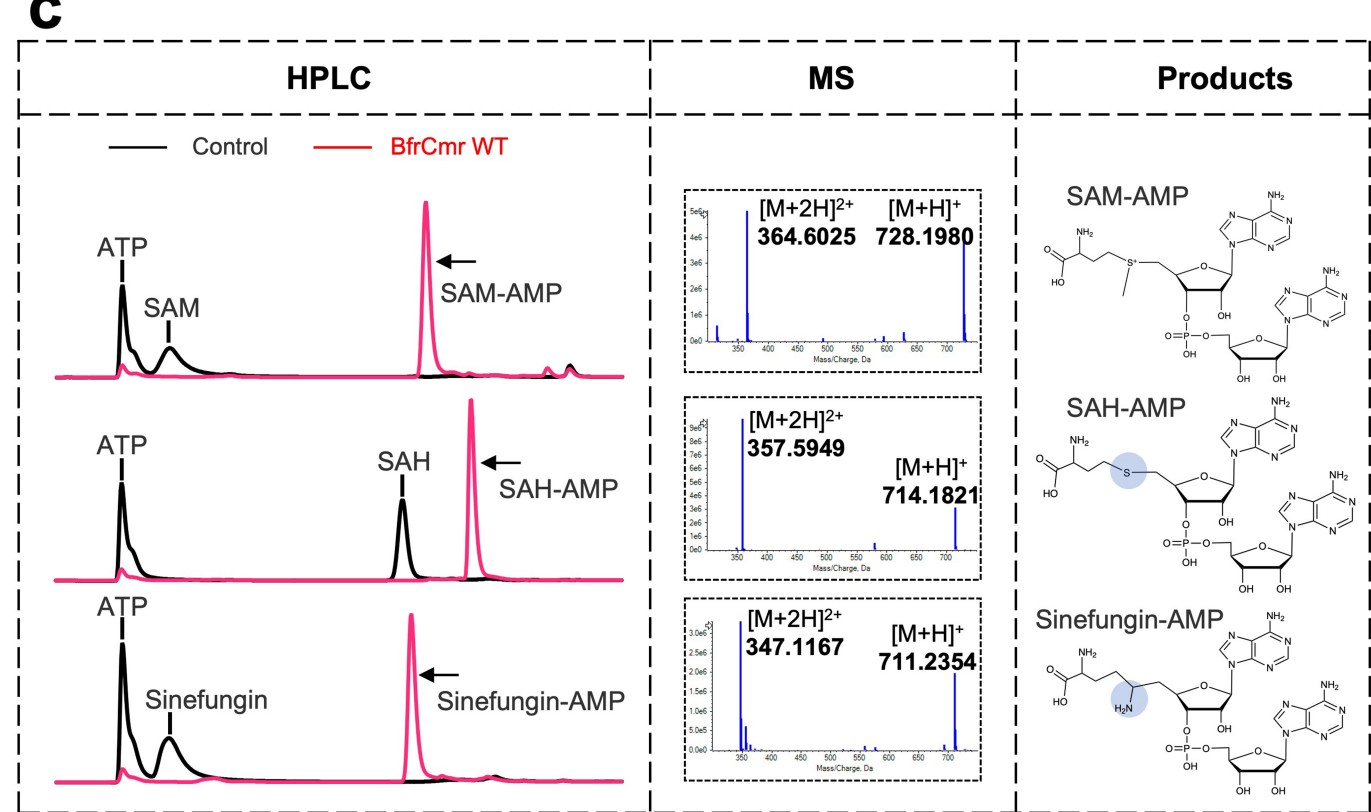

**Extended Data Fig. 4 | In vitro characterisation of *B. fragilis* Cmr-catalysed reaction. a**, ATP (0.5 mM) and SAM (0.5 mM) were incubated with wild type of Cmr (3 μM) in presence of $Mn^{2+}$. Samples were collected at the indicated time points and analysed by HPLC. Cmr was omitted in control samples. **b**, as for **a**, with SAH in place of SAM. **c**, HPLC and MS analysis of Cmr reaction products SAM-AMP, SAH-AMP and sinefungin-AMP. Uncropped HPLC traces are available in Supplementary Data Fig. 6.

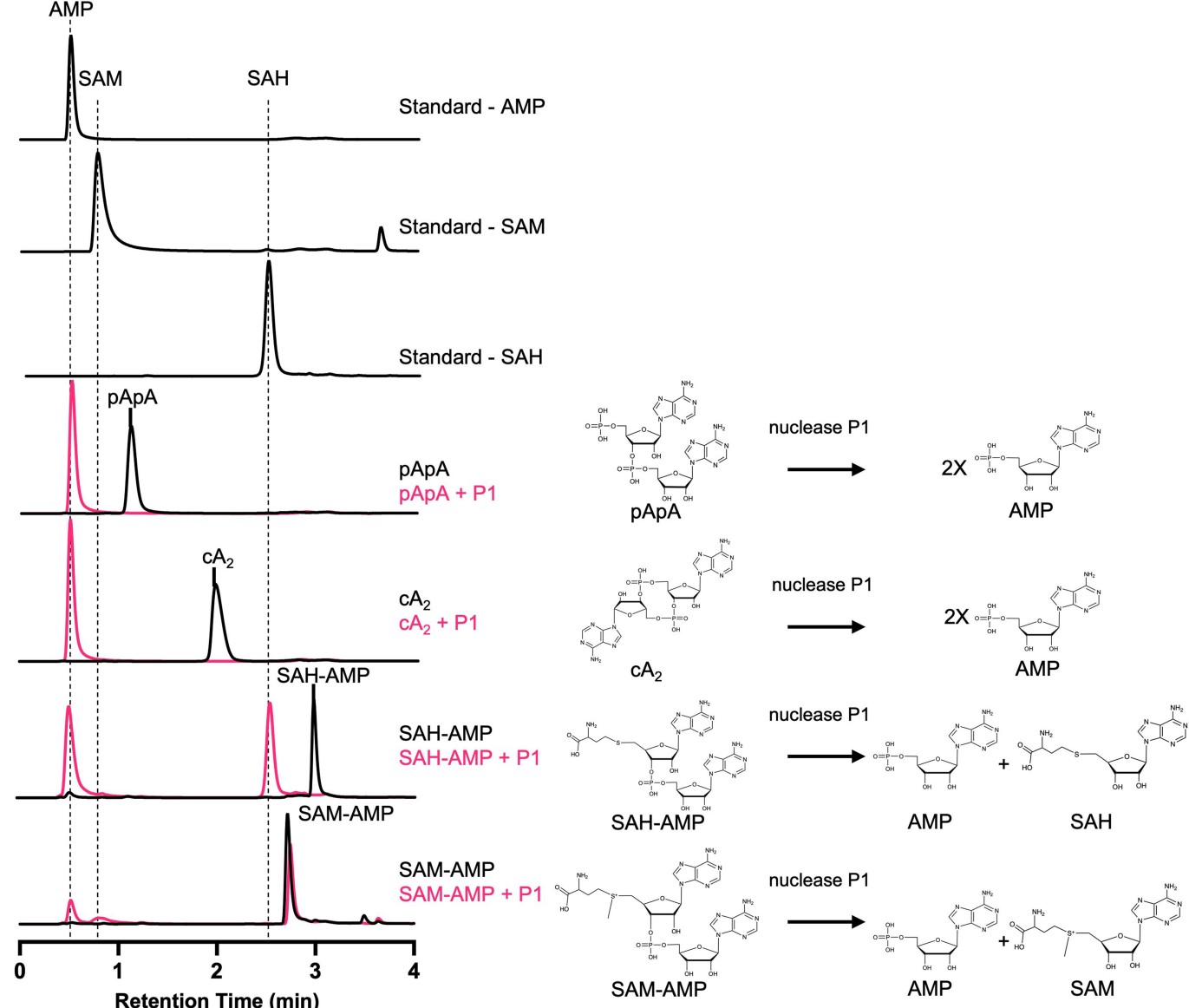

**Extended Data Fig. 5 | P1 nuclease degradation of reaction products.** HPLC analysis of nuclease P1-mediated hydrolysis reactions towards Cmr's products (left). cA$_2$ (cyclic di-3′,5′-adenylate) and pApA (5′- phosphoadenylyl- (3′ → 5′)- adenosine) were used as controls. The proposed reactions of P1 were shown on the right. Uncropped HPLC traces are available in Supplementary Data Fig. 6.

```
              1                                                                              
Bacteroides         MKYIAITLGPITRTIEMAESTKELWAASYFFSYLAKKIVEPFVKKNRT-------------FQL   51
Clostridium      1  MTKNKKSYIGLTIGPIVETIGNAKQTGELWASSYLFSYIMKNIIKTLIVKDREVNKDREQNEKLENRFIL   70
Prevotella       1  MSENKKQYIGITLGPIGRIMSYTKSMKSFWAASYFMSYIGKSLVTDVYKAGRK---------------FLK   56
Camphylobacter   1   MTYIALTIGPIYKTLSNAKKTRELWGGSYLFSYIMKQII--LEFKERE-------------FIV   49
Aliarcobacter    1   MKYIALTIGPIYKTLNSKKTRELWGGSYIFSYIMKQII--LKFEDRD-------------FVT   49
Syntrophothermus 1   MTWFFGITIGPIYGTMSLARDTGGLWAASYLFSYISHCLVQRLREFPRV---------------S   50
Methanococcus    1  MTETNNYLGFTIGPIVQTIASAKKTGHAWGGSYIFSYIMKQMILKLKENNFE-------------ILT   55

                                                    *D70
Bacteroides     52  PLINE------------------EMQKPHCGAGLFPDRYIFK-----SEPGDLELLKQHSDQVLIEIA   96
Clostridium     71  PCIDDILNKLSKNEMEQKSIQKIENNTDKAEQEVGLFHDRFIFQ-----SENGEFDLVKESINEVIDGIV  135
Prevotella      57  PLLSD-------------------KMFDIKDGVGRFPDQYVFI-----AEKDDYEGLLRKRDEVFHTLA  101
Camphylobacter  50  PYIKD-------------------EGIFNAGNEVGLFHDRFIFE-----AQEGDKEALYLVVDEVLISLS   95
Aliarcobacter   50  PYIKD-------------------KTVFKSGNEVGLFHDRFIFT-----SQDGDKQKVQIVIDEVLKELS   95
Syntrophothermus 51 VFSPN------------------PEANRINKRVGAFSDRIYFAYDGDPEADGFFEMLERVKNEAVEEIV  101
Methanococcus   56  PYVEN-------------------EFETEKYGAGLFPDRCIVKVSNDFDKEKILSYFEKIKKEVLNDLA  105

Bacteroides     97  GHIAS--PSLPGTAKDVS-------QIYH-YLKS-YIKIYF------IERTLESDDPHVVIPACEKYLNI  149
Clostridium    136  TGIENHDVCKDKSREEIK-------SDIEDFLKIYYLEIDIDLNQEEKENKLSSENN--IIFKVNRALDA  196
Prevotella     102  IDIAA-----TLNGKYGK-------ADICQYLKE-TIKIYI-----CEINDMKPDEN--AVDKCQGYLAG  151
Camphylobacter  96  EDIKI-------------------DFE-FLKS-YLQINI------VEKELKEGAN--PILELTPYLDM  134
Aliarcobacter   96  NNTRL-------------------SYE-FIKA-YFQINF------IEAEVK--DN--PIIELTPYLDS  132
Syntrophothermus 102 DDIWNSLRSKTHSDQLLK-------EEVRGFLRQ-YLQVYY-----VAIDDSCLYGEP-VLKVLNKLLDA  157
Methanococcus  106  TKIIDETYIKPLDDKESETYDISKLKEFKENLID-YFQIYG------FIGEFSSKKE--AMNTISNYLDS  166

                        *E151
Bacteroides    150  IENQETFPEQEETMISHQKSDFLKFLITNVNGKIYRKDKNSIPRFTGSFLTRDAFGDMN--GERLFESIL  217
Clostridium    197  LELREK--------ILGNNGVENNYILQTLNNKQLKEQD-----FKKHFLSKDAYGENG--KDGDYPSLF  251
Prevotella     152  MECMDIY-------PVTEEHNYLAEYFENINKSC----------LLIKDAFGIGNAKGERIFDTLI  200
Camphylobacter 135  AELFLT--------VSQYQENPLAKVLKGNNS-----------FLTKDAFGE----FLTKDAFGE----KKSFPSLP  175
Aliarcobacter  133  IEQFYK--------IGHYKKNELSKMLKTDNS-------------FLTKDAFGE----RKTFPSLP  173
Syntrophothermus 158 AELRPPY-------VFKNESPHL--MARFLNNETVKKS---------RMFAGAFGV----GKFVFPSLT  204
Methanococcus  167  LELEPNY-------QSHNPCNIKNPLFKFFENKTIKNS---------FLSIDCFETSA--DKEMIKSLP  217

Bacteroides    218  EISASELN---INIQQKALEVITANEKNKGEKYSDQIWDAEEIILNDNKAQLRPYHKYIAIIKSDGDSMG  284
Clostridium    252  KIALDETY---NNEFPNEDEKIKEELKQMGE---------------RKRLKANE-YVAIIQADGDSMG  300
Prevotella     201  EMSANEAV---EKQLITIEDLLLDSSK--------------------VSELMPRYKYVAYISADGDNVG  246
Camphylobacter 176  LIALHDML---KEKPEIKALLNYDEE--------ESVYENRDY-------DFRNYHKYYAIVHADGDSMG  227
Aliarcobacter  174  EIALYDL-----KEHINIKELLKDDEL--------EIYDNKEL-----KKYLKPYHKYIAIVHADGDSMS  225
Syntrophothermus 205 DIAKTETTRRKEKDGKSILEVVVQKLY-AGQSPNSDFCTITNTNTKDPKPRMPKFANYYALVQADGDGIG  273
Methanococcus  218  EIALGNFIESNDIKIDNLNDSVNNGKY-------NEIYKKCAEN----KENFKKHYKYYATIRADGDSVG  276

                                                          **D328-9
Bacteroides    285  ETIKSM---GAYN------IPITQLSKALLSFNIESINEIVAYGGKPIFIGGDDLLCFAPVCCNG-----  340
Clostridium    301  RVIEKF---KIHNDRENIYVDYKDFSSKLLEYDKNSHEKIKKYGGFTIYAGGDDLLFIAPVITKNK----  363
Prevotella     247  KAISKL---G-----------TELSSRLLSFNIKRESVENAGGRLIYAGGDDILFIAPV--------  292
Camphylobacter 228  KVVESL---KSKE-------DFQGFSKKLFDYCSASHDIIKSYGGETIFAGGDDLLFFAPVVSGE-----  282
Aliarcobacter  226  EVI------KDTS-------KLQETSKKLFEYCIHSHKLISDFGGQTIFAGGDDLLFFAPVVSKN-----  277
Syntrophothermus 274 GHITSL---ADIG-------EISNTSRTLLEYADSAVGIMTQYGAFPVYAGGDDLMFFAPLMGWCDGGHP  333
Methanococcus  277  KILKNFLRETDDNAENNNLEKYNIFSKQLFNFSKEVVNIIKEYGGLPIYIGGDDILAFVPLIHENETENE  346

Bacteroides    341  ----NNVFNLVEKLSTCFDQCINQHLQQYINACSEAQRPLPSLSFGISITYHKYPMFEALHTTDYLLMVA  406
Clostridium    364  ----NNIFQLIDVLSGLFDNEFKH-----------EEEKPTTSFGIAIVHHKFPLYYALDE-------A  410
Prevotella     293  ----SSVFQLIKDTDEAFNNEMEQIKDMLKQN----GLDVPTLSYGVSIAYYKHPMREAMEL-------S  347
Camphylobacter 283  ----NSIFSLCDEISQDFDKRF---------------EGTDATLSFGISIQYHKFPLYEALEK-------S  327
Aliarcobacter  278  ----KTIFELLENISIEFNDKF---------------KPKATLSFGVSITYYKFPLYEALEK-------S  321
Syntrophothermus 334 ----VLILNMLRTLDEEFKKRL---------------GDRFSLSFGLAVAYHKYPLYEALEI-------A  377
Methanococcus  347  KDKFKTVFGLIENIDEKFNKEIGKTIKIDG------KEIKPSLSYGLTINYYKYPLYEALNK-------S  403

Bacteroides    407  KDNLFKYTLSNKNIL-NENMKRFILKNKLAFSLQKHSGQIYHTAMSKKGKS-YVKFNMLLQ--KYILKNK  472
Clostridium    411  RDLLFNKAKNYKFNG-EE-------KNAIAFVIKKSGQSFETVVGKKSES-YKQFKELFGNVSNSLNGK  471
Prevotella     348  EELL-GKAKDSG----------RNRICWNMRKHSGQSVGSVFAKENMDVFNKALEIMS--FFGYTGD  401
Camphylobacter 328  RTLLFGKAKSGE-----------KNNIAFSVTKHSGQTFGSVIHKGHKELYENFKLFSS---NIFGGE  381
Aliarcobacter  322  RDLLFTKAKKLP-----------KNNIAFVSGVTHSGQTFGGIIHKSSNA-YEKFLDFV----SIDKSL  373
Syntrophothermus 378 KAQL-DKAKRYKT----------KNALGISLTVHSGQSAQLLLTMNNTLLLDLIETLC----NPLTEE  430
Methanococcus  404  NDLLNYVAKKNKELLTSEADKKEYEKNAIALRILKHSGQEIQITLNKNNKSLYDLFAKLMN--DIILKNN  471

Bacteroides    473  DMSKTQESEKFLSSVIQMIRAHAEILQII-------LQNEDKRTEMLKNYFDNNFNESCHLG--YTGLF  532
Clostridium    472  D----RKIKNYLKAIHFKLKRDKVILNKI-------GKNEEL----LENYFKNNFDEKFHENGDIKKYI  525
Prevotella     402  N-------GLFLHSFSHYLLLHKDMISDV-------LANENSKQQ-LENYIKATFDDDSHTD--HTEII  453
Camphylobacter 382  G-------IDNFLHSLHHKIESNKVVLAEI-------SSSKSK----LQNFFDNYFNKEVHDEL-YREFF  432
Aliarcobacter  374  D-------DNFLHSLHHKIDLHKITIEII-------KSDVAK----LQNFFKNYFNEAGHKE--YEEFF  422
Syntrophothermus 431 N-------VDFLRSVERKLRQRRAVLLQIL------ASNSREERSSRLSWWFSQEFEG------HGATL  480
Methanococcus  472  N--EINENGKILQSFKIKIAEDEVLLNSLYDLHINNGNKDFEEFNTSIDNYFKNNFKKDIHKA--NEQQI  537

Bacteroides    533  EDIQTLLCLRYQENIQDYQNRNEIIQQNTILTSDEKEILIVSPAMDAIHTIFTALQFIHFINYNKDE---  599
Clostridium    526  NYLIKFIYLIYGE-----------IKTDEDKN--------KCIEQIYTYLRFIKFMDEDINLDN-  570
Prevotella     454  KKFHEFMIISSCT-----------YPDGNKG---------KSIELLHALLRYVELIISKN-----  493
Camphylobacter 433  EQLVDFMVEAYKH-------------------EDKE--------KALDVIYSTLRFIKFVQGDKA----  470
Aliarcobacter  423  EKLISYIKEE----------------------QNINNIYGTLRFVKFIKGNKQ----  453
Syntrophothermus 481 KLVQDILGKNLDE-----------------FSSPH-------EALETTEAVLSFARFMGEWEEATQD  523
Methanococcus  538  DNIKNLLKACYYE-------------------YPYDNSKDDNTNNN-TAIQTFERILRLLSFLNEGGE----  585
```

**Extended Data Fig. 6 | Alignment of Cas10 sequences from the CorA-1 cluster.** Key binding site residues shown in Fig. 3 are indicated. Sequence IDs are: *Bacteroides fragilis* ANQ60746.1; *Clostridium botulinum* WP_011986674; *Prevotella - Xylanibacter muris* WP_172276208; *Camphylobacterales* bacterium HIP52383.1; *Aliarcobacter butzleri* WP_260918755; *Syntrophothermus lipocalidus* WP_013175521; *Methanococcus voltae* WP_209917301).

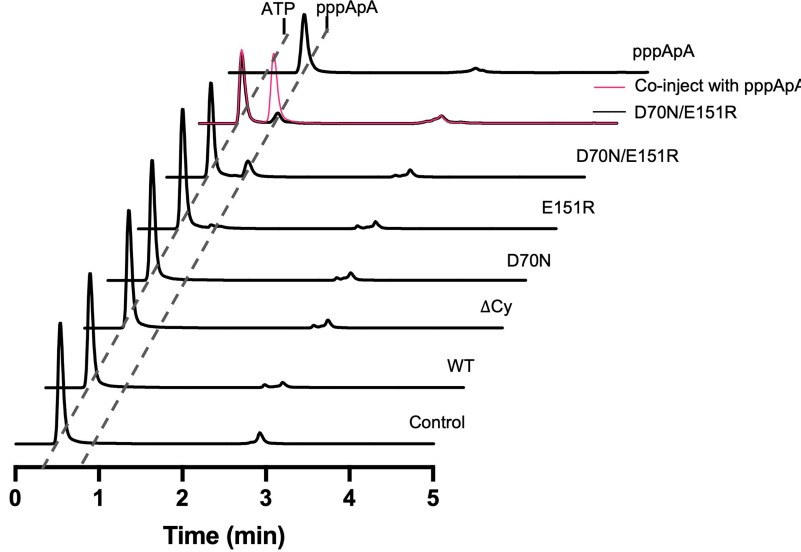

**Extended Data Fig. 7 | pppApA synthesis by the wild-type and variant BfrCmr complexes.** In vitro pppApA synthase activity of wt and variants *B. fragilis* Cmr, analysed by HPLC following incubation of 2 μM Cmr with 0.5 mM ATP for 30 min. The D70N/E151R double mutant synthesises pppApA but not SAM-AMP. Uncropped HPLC traces are available in Supplementary Data Fig. 7.

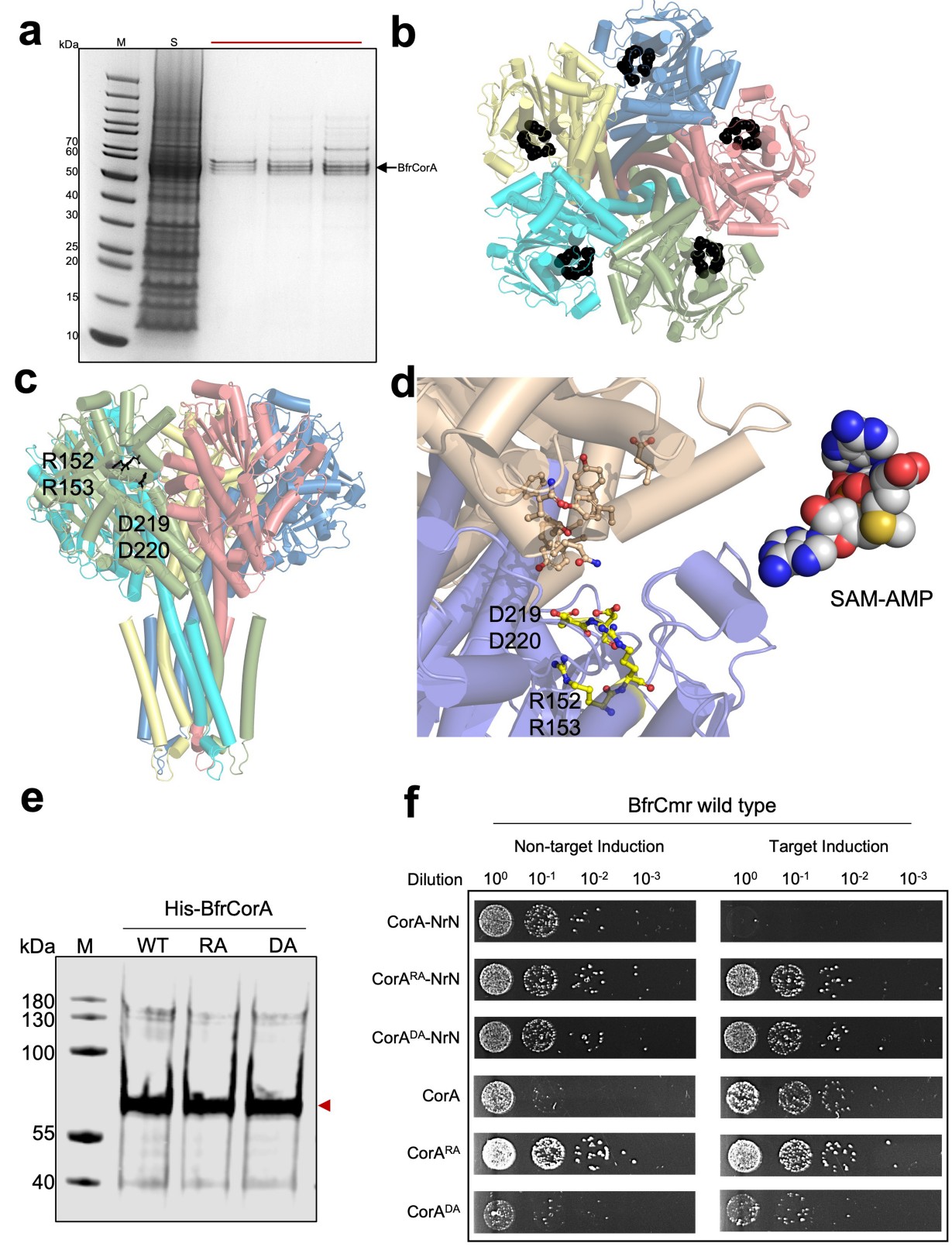

**Extended Data Fig. 8** | See next page for caption.

**Extended Data Fig. 8 | Purification, modelling and mutagenesis of the CorA protein. a**, SDS-PAGE analysis of the final gel filtration step for purification of *B. fragilis* CorA, which was used for SAM-AMP binding assays. The four tightly spaced protein bands in the gel all correspond to CorA, perhaps indicating limited proteolysis of the termini. M – M.Wt. markers, S – sample applied to gel filtration; the horizontal bar indicates the three fractions pooled for further analysis. **b**, Top-down view of the pentameric BfrCorA model with individual subunits coloured differently and the conserved R152/R152/D219/D220 residues indicated by black spheres. **c**, Orthogonal view of the BfrCorA model showing the TM helical bundle at the bottom. **d**, Close up of the intersubunit interface for CorA, with conserved residues (defined in Supplementary Data Fig. 2) shown and a model of SAM-AMP in sphere representation included for scale. **e**, Western blot using the V5 antibody to detect expression of the wild-type (WT), R152A/R153A (RA) and D219A/D220A (DA) variants in *E. coli*. **f**, Plasmid challenge assay, showing that wild-type CorA in conjunction with NrN provides immunity from plasmids carrying a target sequence, but neither CorA variant does. Uncropped images and gels are available in Supplementary Data Fig. 8.

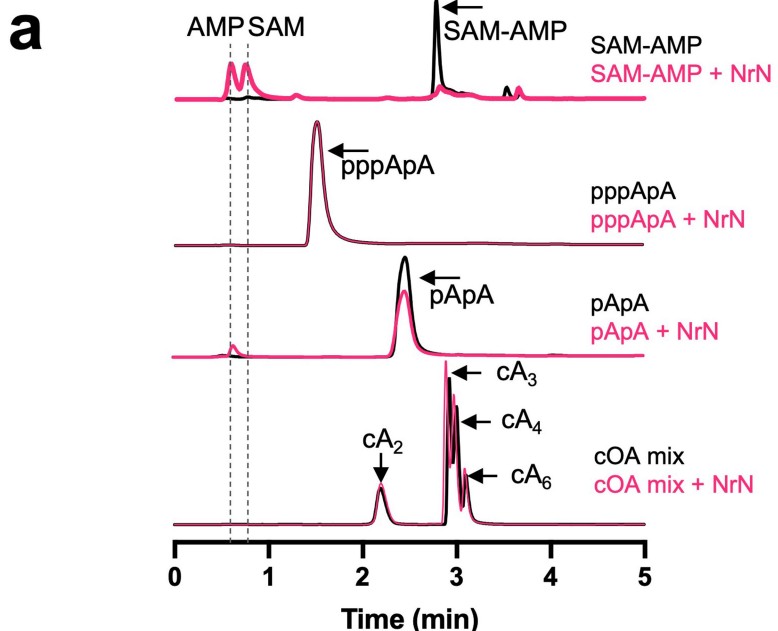

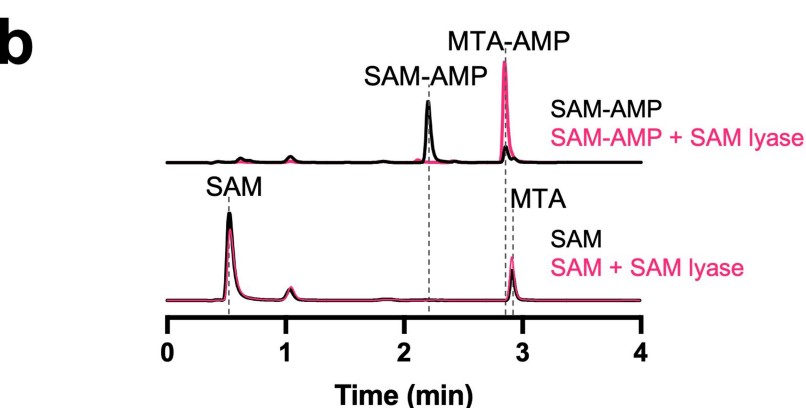

**Extended Data Fig. 9 | Specificity of the NrN and SAM-AMP lyase effectors.**
**a**, HPLC analysis of reaction products when the NrN PDE was incubated with
$cA_2$, $cA_3$, $cA_4$, $cA_6$, pApA and SAM-AMP for 30 min. Only SAM-AMP is a substrate
for NrN. **b,** HPLC analysis of reaction products for SAM-AMP and SAM following
incubation with *C. botulinum* SAM-AMP lyase for 30 min. SAM-AMP is preferred
as a substrate over SAM. Uncropped gel and HPLC traces are available in
Supplementary Data Fig. 9.

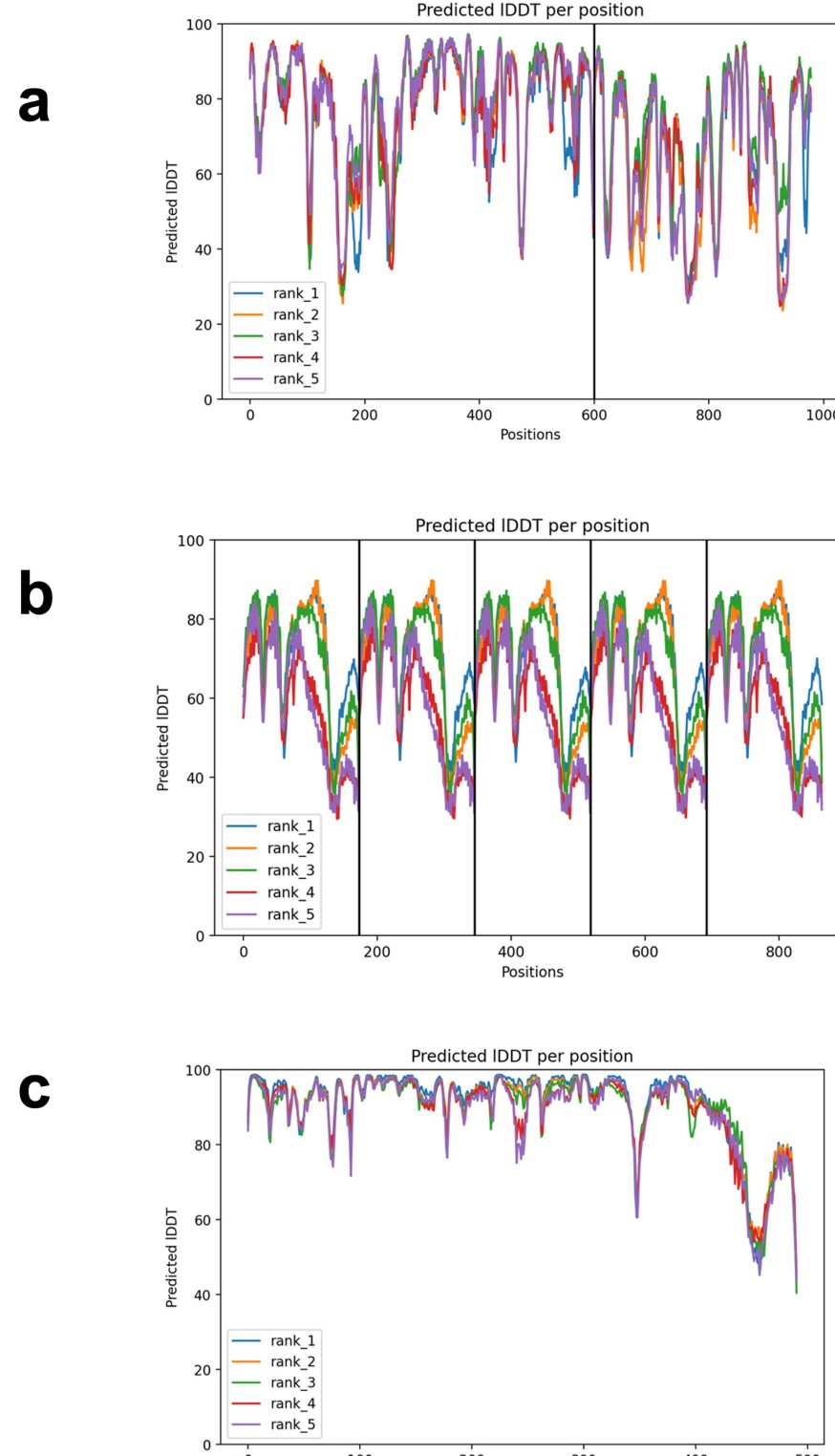

**Extended Data Fig. 10 | Local distance difference test (lDDT) scores for Alphafold2 models. a**, BfrCas10–5 complex; **b**, pentameric BfrCorA C-terminal TM domain; **c**, monomeric full-length BfrCorA protein.

# Reporting Summary

## Statistics

For all statistical analyses, confirm that the following items are present in the figure legend, table legend, main text, or Methods section.

| n/a | Confirmed | |
|---|---|---|
| ☐ | ☒ | The exact sample size (*n*) for each experimental group/condition, given as a discrete number and unit of measurement |
| ☐ | ☒ | A statement on whether measurements were taken from distinct samples or whether the same sample was measured repeatedly |
| ☒ | ☐ | The statistical test(s) used AND whether they are one- or two-sided *Only common tests should be described solely by name; describe more complex techniques in the Methods section.* |
| ☒ | ☐ | A description of all covariates tested |
| ☒ | ☐ | A description of any assumptions or corrections, such as tests of normality and adjustment for multiple comparisons |
| ☐ | ☒ | A full description of the statistical parameters including central tendency (e.g. means) or other basic estimates (e.g. regression coefficient) AND variation (e.g. standard deviation) or associated estimates of uncertainty (e.g. confidence intervals) |
| ☒ | ☐ | For null hypothesis testing, the test statistic (e.g. *F*, *t*, *r*) with confidence intervals, effect sizes, degrees of freedom and *P* value noted *Give P values as exact values whenever suitable.* |
| ☒ | ☐ | For Bayesian analysis, information on the choice of priors and Markov chain Monte Carlo settings |
| ☒ | ☐ | For hierarchical and complex designs, identification of the appropriate level for tests and full reporting of outcomes |
| ☒ | ☐ | Estimates of effect sizes (e.g. Cohen's *d*, Pearson's *r*), indicating how they were calculated |

*Our web collection on statistics for biologists contains articles on many of the points above.*

## Software and code

Policy information about availability of computer code

| Data collection | Phosphorimage data were collected using GE Healthcare ImageQuant TL 7.0 <br> HPLC data were collected using Chromeleon 6.8 Chromatography Data System software (ThermoFisher). |
|---|---|
| Data analysis | Graphpad Prism 9 was used for data analysis. Bioinformatic analysis was carried out using a custom script. These steps were wrapped in a Snakemake 7.22.0 48 pipeline and an R script (Rstudio 2021.9.0.351) that are available in Github: https://github.com/vihoikka/Cas10_prober. Software packages used in the pipeline were Hmmer 3.3.2, CD-hit 4.8.1, CRISPRCasTyper 1.8.0, Muscle 5.1, FastTree 2.1.11, R4.1.1). Plasmid maps were generated using freeware Benchling software (https://benchling.com/) |

For manuscripts utilizing custom algorithms or software that are central to the research but not yet described in published literature, software must be made available to editors and reviewers. We strongly encourage code deposition in a community repository (e.g. GitHub). See the Nature Portfolio guidelines for submitting code & software for further information.

## Data

Policy information about availability of data

 All manuscripts must include a data availability statement. This statement should provide the following information, where applicable:

- - Accession codes, unique identifiers, or web links for publicly available datasets
- - A description of any restrictions on data availability
- - For clinical datasets or third party data, please ensure that the statement adheres to our policy

> The genome sequences used to make figure 1a were downloaded from NCBI (https://www.ncbi.nlm.nih.gov) on 14/03/2023.
> Mass spectrometry data are available on FigShare at the following address: https://doi.org/10.6084/m9.figshare.c.6646859.v1
> Experimentally determined 3D protein structure coordinates are available at https://www.rcsb.org

## Research involving human participants, their data, or biological material

Policy information about studies with human participants or human data. See also policy information about sex, gender (identity/presentation), and sexual orientation and race, ethnicity and racism.

| | |
|---|---|
| Reporting on sex and gender | not applicable |
| Reporting on race, ethnicity, or other socially relevant groupings | not applicable |
| Population characteristics | not applicable |
| Recruitment | not applicable |
| Ethics oversight | not applicable |

Note that full information on the approval of the study protocol must also be provided in the manuscript.

# Field-specific reporting

Please select the one below that is the best fit for your research. If you are not sure, read the appropriate sections before making your selection.

☒ Life sciences  ☐ Behavioural & social sciences  ☐ Ecological, evolutionary & environmental sciences

For a reference copy of the document with all sections, see nature.com/documents/nr-reporting-summary-flat.pdf

# Life sciences study design

All studies must disclose on these points even when the disclosure is negative.

| | |
|---|---|
| Sample size | technical triplicates and biological duplicates or triplicates (as stated) were obtained for all measurements. This represents the accepted norm for a biochemical study, allowing standard deviation and mean to be calculated. |
| Data exclusions | No data were excluded |
| Replication | Duplicate or triplicate experiments were carried out, as stated for each experiment |
| Randomization | Randomization was not applied to the small biochemical  datasets studied, in keeping with established norms for this type of study. |
| Blinding | Blinding  was not applied to the small biochemical  datasets studied, in keeping with established norms for this type of study. |

# Reporting for specific materials, systems and methods

We require information from authors about some types of materials, experimental systems and methods used in many studies. Here, indicate whether each material, system or method listed is relevant to your study. If you are not sure if a list item applies to your research, read the appropriate section before selecting a response.

## Materials & experimental systems

| n/a | Involved in the study |
|---|---|
| ☐ | ☒ Antibodies |
| ☒ | ☐ Eukaryotic cell lines |
| ☒ | ☐ Palaeontology and archaeology |
| ☒ | ☐ Animals and other organisms |
| ☒ | ☐ Clinical data |
| ☒ | ☐ Dual use research of concern |
| ☒ | ☐ Plants |

## Methods

| n/a | Involved in the study |
|---|---|
| ☒ | ☐ ChIP-seq |
| ☒ | ☐ Flow cytometry |
| ☒ | ☐ MRI-based neuroimaging |

## Antibodies

| | |
|---|---|
| Antibodies used | V5 tag monoclonal antibody was purchased from Thermofisher (cat. no. R960-25; clone SV5-Pk1)<br>anti-mouse IgG was purchased from LI-COR Biosciences, catalog: 926-32212, Lot No. D10811-15 |
| Validation | Western blot analysis of V5 tag was performed by loading 20 μg of whole cell extracts of untransfected HEK-293 and HEK-293 transiently overexpressing V5- His-LacZ using Novex® NuPAGE® 4-12% Bis-Tris gel (Product # NP0321BOX), Xcell SureLock Electrophoresis system (Product # EI0002), Novex sharp Pre-stained Protein Standard (LC5800). Proteins were transferred to a PVDF membrane and blocked with 5% skim milk for 1 hour at room temperature. V5- His-LacZ was detected at ~117 kDa using V5 Tag Mouse Monoclonal Antibody (Product # R960-25) at a 1:1000 dilution in 2.5% skim milk at 4°C overnight on a rocking platform. Goat Anti-Mouse IgG - HRP Secondary Antibody (Product # 62-6520) at 1:4000 dilution was used and chemiluminescent detection was performed using Pierce™ ECL Western Blotting Substrate (Product # 32106).<br>The anti-mouse IgG antibody was isolated by affinity chromatography using antigens coupled to agarose beads. Based on ELISA, this antibody reacts with the heavy and light chains of mouse IgG and with the light chains of mouse IgM and IgA. This antibody was tested by ELISA and/or solid-phase adsorbed to ensure minimal cross-reaction with bovine, chicken, goat, guinea pig, horse, human, rabbit, and sheep serum proteins, but the antibody may cross-react with immuno-globulins from other species. The conjugate has been specifically tested and qualified for Western blot and In-Cell Western Assay applications. |

