## [Peer Review File · Nature]

Manuscript Title: Antiviral Type III CRISPR signalling via conjugation of ATP and AdoMet

Reviewer Comments & Author Rebuttals

Reviewer Reports on the Initial Version:

Referee expertise:

Referee #1: evolution of CRISPR-Cas, bioinformatics, phage defence systems

Referee #2: biochemistry and structural bio of phage defence systems

Referee #3: molecular mechanisms of CRISPR, structural bio

Referees' comments:

Referee #1 (Remarks to the Author):

This a great, groundbreaking paper that reports the discovery of a novel second messenger, SAM-AMP, that is synthesized by some variants of Cas10 (type III CRISPR effector) and activates the associated membrane protein CorA, apparently, to trigger bacterial cell death via membrane permeabilization. The authors also demonstrate that the NrN nuclease or alternatively SAM lyase associated with the respective CRISPR systems hydrolyze SAM-AMP and function as off-switches for the SAM-AMP mediated cell death pathway. These results are remarkable at multiple levels including the discovery of the new signaling molecule as such, the demonstration of the striking enzymatic flexibility of the Cas10 cyclase and elucidation of the function of the prominent but previously enigmatic membrane association of type III CRIPSR systems. The experiments presented in the paper are convincing.

There are obvious enough limitations, above all, the fact that all the experiments are done in a model rather than the native system, a typical objection to this type of work. Obviously, validating these results in *B. fragilis* would be great. However, I believe it would be utterly disingenuous to pose this as a criticism. It is quite obvious that, at best, it would take years to develop that native system if feasible at all. Further, the function of CorA is not well characterized, and it is unclear how exactly does SAM-AMP makes CorA into a pore. Structures of CorA, bound to SAM-AMP vs free, certainly, will help. However, I am convinced that this is out of the scope of the present work and is the direction of further experimentation, by the authors and/or others. Perhaps, it would be useful to be more explicit in the Discussion about these (fully understandable) limitations of the present study and future directions

Other than the possible minor extension of the Discussion mentioned above, I see no problems with this manuscript and consider it ready for prime time.

Very minor points

This "As described previously, most type III systems with a CorA effector also encode a PDE of the NrN or DEDD family" requires a reference - ref. 16, already cited elsewhere.

Here "Immunity was lost when a variant of NrN with mutation of the DHH active site motif (D85A:H86A:H87A).." something seems to be missing ("was included in the reaction" or some such).

Referee #2 (Remarks to the Author):

Chi et al present the discovery of "SAM-AMP" as a bacterial nucleotide second messenger functioning in Type III CRISPR defense. The experiments include an exceptional series of in vivo, biochemical, bioinformatic, and structural modeling analyses that explain a near complete model of SAM-AMP synthesis, effector function, and degradation during multiple models of bacterial plasmid resistance. Notably, SAM-AMP represents an entirely new class of nucleotide second messengers. The results are incredibly exciting and are certain to have an immediate wide impact on multiple related fields.

I have a suggested control experiment to help further verify the chemical structure of SAM-AMP. Otherwise, I have only minor comments to help improve clarity of the manuscript for a general audience.

1) The authors present the chemical structure of SAM-AMP as 3'-adenylyl-AdoMet. An overall convincing argument for this structure is based on HPLC, MS2 fragmentation, incorporation of SAM-analogs, and partial susceptibility to degradation with nuclease P1. However, a concern is that identification of the proposed 3'-5'-linkage is based primarily on degradation with nuclease P1, but the data in Extended Data Figure 5 indicate that SAM-AMP is only partially susceptible to P1 cleavage with a majority of the signal unchanged upon treatment with this nuclease. It's not clear that these data rule out a 2'-5' bond, a more complex linkage, or a mixed population of potential signaling molecules in the sample.

Are the authors able to compare enzymatically synthesized SAM-AMP with a chemical standard for 3'-adenylyl-AdoMet? Alternatively, do the authors have enough material for NMR analysis? The overall data presented in the manuscript are strong, but the nucleotide second messenger field has been surprised by chemical linkage several times before.

2) The plasmid resistance assays in Figure 1 are confusing to me. A few related points that would be useful to clarify in the text:

a. Is it correct that this E. coli assay relies on expression of four different plasmids? My understanding is there is a (1) pBfrCmr1-6 plasmid to express the cmr1-6 proteins, (2) a pBfrCRISPR plasmid to express Cas6 and a CRISPR array, (3) a targeting pCRISPR-Tet plasmid crRNA, and (4) a challenge pRAT plasmid expressing the CorA/NrN effector proteins. It is confusing how such a

complex system works in the cell with origins of replication and it would be helpful to include a more detailed schematic in Extended Data Figure 1A.

b. Why is plasmid resistance observed for the wildtype / non-Target / CorA experiment in Figure 1? The authors briefly discuss this in the text, but it's not clear why CorA-toxicity would change with target or non-target crRNA?

c. If NrN is the off switch, why is it needed for plasmid resistance? Do NrN and CorA interact or does NrN impact stability of CorA? The authors discuss this in the text but the explanation doesn't seem to agree with the overall model of signaling.

3) Text Corrections

a. Line 120 is missing a closed parentheses symbol.

b. The resolution of the plasmid map images in ED1A is quite low.

c. Discussion of the position of site 1 cleavage (lines 120–125 and Figures ED3B–D) and the crRNA:target RNA duplex boundary would benefit from additional labels or colors in the figure. Perhaps add an arrow to indicate site 1 in Figure ED3B and label each arrowhead in ED3A with a name (e.g. site 1, site 2, etc.) to clarify the text and figures.

I hope the authors will find my comments useful, thank you for the opportunity to read this exciting manuscript.

Philip Kranzusch

Referee #3 (Remarks to the Author):

This manuscript reports the first identification of a novel signaling molecule SAM-AMP (3'-adenylyl-AdoMet) that is produced by the Type III CRISPR-Cas system in response to MGE infection. It has been shown before that the Cas10 protein of the Type III CRISPR-Cas system is able to produce cyclic oligoadenylate (cAn) molecules from ATP. Authors show here that instead of cAn, the Cas10 protein of the *B.fragilis* Type III CRISPR-Cas system produces SAM-AMP (3'-adenylyl-AdoMet) conjugating ATP to S-adenosyl methionine via a phosphodiester bond. SAM-AMP acts a signaling molecule that activates CRISPR-Cas associated protein CorA, presumably triggering cell death. Authors also show that SAM-AMP signal is de-activated by CRISPR associated phosphodiesterases or a SAM-AMP lyase. The manuscript is technically sound, well written and describes unexpected findings that are of interest for the general journal readership.

Comments/observations are:

1. Line 9-13: authors did not mention RNase activity that also directly contributes to the Type III system defense against MGEs.
2. Line 53: Fig. 1C. Schematic representation of the plasmid challenge assay would be helpful.
3. Lines 88-90: Based on their experimental data (Fig. 1) authors claim that CorA expression is toxic to the cells. On the other hand, co-expression of CorA together with Nrn seems to neutralize CorA

toxicity. Do they form a complex? The CorA toxicity seems to be also decreased after deletion of the transmembrane helices. However, I find it puzzling that no CorA toxicity is manifested in the background of the Mtb system. Is the expression level of CorA similar in both cases? Overexpression of CorA may cause the toxicity and it would be interesting to see whether Nrn neutralizes it through the complex formation.

4. Lines 117-119: Authors claim that “5'-end labelled target RNA was cleaved at four positions with 6 nt spacing, corresponding to the placement of the Cas7 active sites in the backbone”. How many Cmr4 subunits are present in the complex? Does the number of subunits in the complex correlate with the number of cleavage sites (Extended Figure 3)? Why there is no cleavage between 4 nt and 13 nt? Are some Cmr4 subunits not cleaving or misfiring?

5. Authors show (Fig. 4) that CorA binds SAM-AMP? Where does the SAM-AMP binding site resides? Do CorA sequence analysis or AF models help to predict it?

Author Rebuttals to Initial Comments:

Response to Referees:

Referee #1 (Remarks to the Author):

This a great, groundbreaking paper that reports the discovery of a novel second messenger, SAM-AMP, that is synthesized by some variants of Cas10 (type III CRISPR effector) and activates the associated membrane protein CorA, apparently, to trigger bacterial cell death via membrane permeabilization. The authors also demonstrate that the NrN nuclease or alternatively SAM lyase associated with the respective CRISPR systems hydrolyze SAM-AMP and function as off-switches for the SAM-AMP mediated cell death pathway. These results are remarkable at multiple levels including the discovery of the new signaling molecule as such, the demonstration of the striking enzymatic flexibility of the Cas10 cyclase and elucidation of the function of the prominent but previously enigmatic membrane association of type III CRISPR systems. The experiments presented in the paper are convincing.

*There are obvious enough limitations, above all, the fact that all the experiments are done in a model rather than the native system, a typical objection to this type of work. Obviously, validating these results in *B. fragilis* would be great. However, I believe it would be utterly disingenuous to pose this as a criticism. It is quite obvious that, at best, it would take years to develop that native system if feasible at all. Further, the function of CorA is not well characterized, and it is unclear how exactly does SAM-AMP makes CorA into a pore. Structures of CorA, bound to SAM-AMP vs free, certainly, will help. However, I am convinced that this is out of the scope of the present work and is the direction of further experimentation, by the authors and/or others. Perhaps, it would be useful to be more explicit in the Discussion about these (fully understandable) limitations of the present study and future directions*

Other than the possible minor extension of the Discussion mentioned above, I see no problems with this manuscript and consider it ready for prime time.

Thanks for these enthusiastic comments. In response to the suggestions from all three referees we have extended the discussion to highlight the ongoing uncertainties in the experimental system.

Very minor points

This "As described previously, most type III systems with a CorA effector also encode a PDE of the NrN or DEDD family" requires a reference - ref. 16, already cited elsewhere.

Reference added.

Here "Immunity was lost when a variant

of NrN with mutation of the DHH active site motif (D85A:H86A:H87A).. " something seems to be missing ("was included in the reaction" or some such).

We corrected this sentence to read: "Immunity was lost when wild-type NrN was substituted with a variant mutated in the DHH active site motif (D85A:H86A:H87A),"

Referee #2 (Remarks to the Author):

Chi et al present the discovery of "SAM-AMP" as a bacterial nucleotide second messenger functioning in Type III CRISPR defense. The experiments include an exceptional series of in vivo, biochemical, bioinformatic, and structural modeling analyses that explain a near complete model of SAM-AMP synthesis, effector function, and degradation during multiple models of bacterial plasmid resistance. Notably, SAM-AMP represents an entirely new class of nucleotide second messengers. The results are incredibly exciting and are certain to have an immediate wide impact on multiple related fields.

I have a suggested control experiment to help further verify the chemical structure of SAM-AMP. Otherwise, I have only minor comments to help improve clarity of the manuscript for a general audience.

1) The authors present the chemical structure of SAM-AMP as 3'-adenylyl-AdoMet. An overall convincing argument for this structure is based on HPLC, MS2 fragmentation, incorporation of SAM-analogs, and partial susceptibility to degradation with nuclease P1. However, a concern is that identification of the proposed 3'-5'-linkage is based primarily on degradation with nuclease P1, but the data in Extended Data Figure 5 indicate that SAM-AMP is only partially susceptible to P1 cleavage with a majority of the signal unchanged upon treatment with this nuclease. It's not clear that these data rule out a 2'-5' bond, a more complex linkage, or a mixed population of potential signaling molecules in the sample.

Are the authors able to compare enzymatically synthesized SAM-AMP with a chemical standard for 3'-adenylyl-AdoMet? Alternatively, do the authors have enough material for NMR analysis? The overall data presented in the manuscript are strong, but the nucleotide second messenger field has been surprised by chemical linkage several times before.

Thanks for this constructive suggestion. We have based our assignment of a 3'-5' phosphodiester bond on two observations: the known specificity of PALM family polymerases and the the fact that SAH-AMP is completely degraded by P1 nuclease (Ext Data Fig. 5). The partial degradation of SAM-AMP may reflect the fact that this molecule is resistant to the wide variety of ribonucleases present in E. coli, which may be relevant for function. Nevertheless, while we might be 90% certain of the 3'-5' bond we are not 100%. We have looked into chemical synthesis of a standard, but have been advised that this is extremely difficult due to the stereochemistry of the sulfur atom. We cannot yet make enough of the molecule for structure determination by NMR. In the circumstances, we have clarified the level of uncertainty that exists for the phosphodiester bond as follows:

“While SAH-AMP was completely degraded by nuclease P1, only partial degradation of SAM-AMP was observed (Extended Data, Fig. 5). Thus, while we consider a 3’-5’ phosphodiester linkage highly likely, we cannot rule out a 2’-5’ linkage completely. Final confirmation of the linkage will require further analysis, for example by NMR.”

We have also removed the specific reference to a 3’-5’ bond in the summary, and altered the relevant text in figure 2 as follows: *“d, The proposed structure of the signal molecule, whose fragmentation pattern is shown by dotted arrows. The MS/MS data cannot distinguish between 2’-5’ and 3’-5’ phosphodiester bonds. The latter is more likely and is shown here, but a 2’-5’ bond cannot be completely ruled out presently.”*

2) The plasmid resistance assays in Figure 1 are confusing to me. A few related points that would be useful to clarify in the text:

a. Is it correct that this E. coli assay relies on expression of four different plasmids? My understanding is there is a (1) pBfrCmr1-6 plasmid to express the cmr1–6 proteins, (2) a pBfrCRISPR plasmid to express Cas6 and a CRISPR array, (3) a targeting pCRISPR-Tet plasmid crRNA, and (4) a challenge pRAT plasmid expressing the CorA/NrN effector proteins. It is confusing how such a complex system works in the cell with origins of replication and it would be helpful to include a more detailed schematic in Extended Data Figure 1A.

There are in fact three plasmids: pBfrCmr1-6, pBfrCRISPR and the pRAT challenge plasmid which contains both the effector gene(s) and the tetR target gene. To clarify, we have added a schematic to Ext data fig 1 as suggested.

b. Why is plasmid resistance observed for the wildtype / non-Target / CorA experiment in Figure 1? The authors briefly discuss this in the text, but it’s not clear why CorA-toxicity would change with target or non-target crRNA?

This is an aspect of the plasmid challenge assay that we currently do not understand, but it is reproducible. We have added discussion of this enigmatic aspect of the system in the Discussion.

c. If NrN is the off switch, why is it needed for plasmid resistance? Do NrN and CorA interact or does NrN impact stability of CorA? The authors discuss this in the text but the explanation doesn’t seem to agree with the overall model of signaling.

We have seen no evidence for the physical interaction of NrN and CorA, although the genes are sometimes fused. The range of partner proteins, which degrade SAM-AMP in different ways, suggest that it is the signal molecule that must be targeted, but as the referee points out this doesn’t explain the strict requirement for this activity for defence. We have put further thought into this,

and although we don't have a definitive answer, one possibility is that high levels of SAM-AMP lead to "desensitisation" of the CorA channel – a phenomenon seen for other ligand-gated ion channels. A molecular explanation for this might be that binding of SAM-AMP to all five binding sites results in channel closure. This is speculation, but could explain the data. We have added further discussion of this point and other points of uncertainty to the discussion, as follows:

"There remain some important open questions. The toxicity of CorA when no SAM-AMP is synthesised and NrN is absent is hard to explain, likewise the observation that both CorA and NrN are required for immunity. These data suggest a close functional link, although we have detected no physical interaction between the two proteins in vitro. Rather than functioning in a manner analogous to ring nucleases, an alternative hypothesis is that NrN (or SAM-AMP lyase) is required to prevent de-sensitization of the CorA channel – a phenomenon observed for other pentameric ligand-gated ion channels when activator concentrations remain high 40. Answers to these questions will likely require further analysis of the system in a cognate host at native expression levels, coupled with structure: function studies of the CorA channel."

3) Text Corrections

- a. Line 120 is missing a closed parentheses symbol. *Corrected*
- b. The resolution of the plasmid map images in ED1A is quite low. *Fixed*
- c. Discussion of the position of site 1 cleavage (lines 120–125 and Figures ED3B–D) and the crRNA:target RNA duplex boundary would benefit from additional labels or colors in the figure. Perhaps add an arrow to indicate site 1 in Figure ED3B and label each arrowhead in ED3A with a name (e.g. site 1, site 2, etc.) to clarify the text and figures. *Fixed*.

I hope the authors will find my comments useful, thank you for the opportunity to read this exciting manuscript.

Philip Kranzusch

Referee #3 (Remarks to the Author):

This manuscript reports the first identification of a novel signaling molecule SAM-AMP (3'-adenylyl-AdoMet) that is produced by the Type III CRISPR CRISPR-Cas system in response to MGE infection. It has been shown before that the Cas10 protein of the Type III CRISPR-Cas system is able to produce cyclic oligoadenylate (cAn) molecules from ATP. Authors show here that instead of cAn, the Cas10 protein of the B.fragilis Type III CRISPR-Cas system produces SAM-AMP (3'-adenylyl-AdoMet) conjugating ATP to S-adenosyl methionine via a phosphodiester bond. SAM-AMP acts a signaling molecule that activates CRISPR-Cas associated protein CorA, presumably triggering cell death. Authors also show that SAM-AMP signal is de-activated by CRISPR associated phosphodiesterases or a SAM-AMP lyase. The manuscript is technically sound, well written and describes unexpected findings that are of interest for the general journal readership.

Comments/observations are:

1. Line 9-13: authors did not mention RNase activity that also directly contributes to the Type III system defense against MGEs.

We have added a sentence and references to address this point at the relevant place in the results: "Type III CRISPR systems also cleave bound target RNA using the Cas7 subunit, either for direct defence against MGE²⁹ or for regulatory purposes⁸."

2. Line 53: Fig. 1C. Schematic representation of the plasmid challenge assay would be helpful.

Now added in Ext Data Figure 1.

3. Lines 88-90: Based on their experimental data (Fig. 1) authors claim that CorA expression is toxic to the cells. On the other hand, co-expression of CorA together with Nrn seems to neutralize CorA toxicity. Do they form a complex? The CorA toxicity seems to be also decreased after deletion of the transmembrane helices. However, I find it puzzling that no CorA toxicity is manifested in the background of the Mtb system. Is the expression level of CorA similar in both cases? Overexpression of CorA may cause the toxicity and it would be interesting to see whether Nrn neutralizes it through the complex formation.

Please see our response to similar comments from referee 2 above.

4. Lines 117-119: Authors claim that "5'-end labelled target RNA was cleaved at four positions with 6 nt spacing, corresponding to the placement of the Cas7 active sites in the backbone". How many Cmr4 subunits are present in the complex? Does the number of subunits in the complex correlate with the number of cleavage sites (Extended Figure 3)? Why there is no cleavage between 4 nt and 13 nt? Are some Cmr4 subunits not cleaving or misfiring?

In the absence of a cryo-EM structure, we do not know for sure how many Cas7 subunits are present in the BfrCmr complex. We note that missed cleavage sites have been observed in the SsoCsm system, and we have included the following sentence: "In the absence of structural data to define the number and positions of Cas7 subunits in the complex, we could not analyse the cleavage pattern further with any degree of certainty".

5. Authors show (Fig. 4) that CorA binds SAM-AMP? Where does the SAM-AMP binding site resides? Do CorA sequence analysis or AF models help to predict it?

In response to this question, we have carried out further modelling studies and added new data to help define the SAM-AMP binding site (New Figure, Ext Data Fig 9). A model for pentameric BfrCorA was built by superimposing the available AF2 model of the BfrCorA monomer on to each of the five transmembrane helical domains of the CorA structure (PDB 2IUB). We next analysed the CorA sequences from the CorA1 clade, generating a multiple sequence alignment to highlight conserved residues (Supplementary Data figure 9). A clear cluster of conserved residues was apparent in the AF2 model at the interfaces between adjacent monomers. Hypothesising that this constitutes the binding site, we generated site directed variants that targeted two pairs of conserved residues, 152RR and 219DD, by converting them to alanine. The resulting variants were tested in our plasmid challenge assay and shown to confer no resistance to plasmid transformation. The relative expression levels of the wild-type and variant BfrCorA proteins were checked by Western blotting to confirm protein stability and expression. This new data and accompanying discussion have been added to the manuscript. The relevant paragraph reads as follows:

“To investigate this in more detail we generated a model of the pentameric CorA structure (Extended Data Fig. 9) and mapped the positions of conserved residues in the CorA1 clade identified from a multiple sequence alignment (Supplementary Data Fig. 9). A cluster of conserved residues at the interdomain interface hinted at a putative SAM-AMP binding site. To test this, we created two site-directed variants of CorA by mutating two pairs of conserved residues (R152/R153 and D219/D220) in this cluster to alanine. The variant proteins were expressed similarly to the wild-type CorA, but no longer provided immunity in the plasmid challenge assay, consistent with a role in SAM-AMP binding (Extended Data Fig. 9).”

Reviewer Reports on the First Revision:

Referees' comments:

Referee #1 (Remarks to the Author):

The authors have addressed my comments (which were anyway minor) and, to the best of my judgment, those of the other reviewers in a thorough and fully satisfactory manner. No further comments.

Referee #2 (Remarks to the Author):

Chi et al's revised manuscript is significantly improved. The authors have addressed each of my points with improved clarity in the text and figures and inclusion of more cautious language describing the phosphodiester linkage of SAM-AMP.

I am less confident than the authors that the linkage of SAM-AMP is a 3'–5' bond. There doesn't appear to be a reason that Cas10/PALM enzymes would be precluded from synthesizing 2'–5' bonds and given the enormous diversity of Type III CRISPR systems it seems more likely than not that some systems rely on alternative phosphodiester linkages. I agree the text changes in the revised manuscript are sufficient and no further experiments are needed for publication, but I do hope the authors continue to study this system as it will be of high interest to the field to understand the specific chemistry of SAM-AMP synthesis. I congratulate the authors on an incredible study, the discovery of SAM-AMP is very exciting!

Philip Kranzusch

Referee #3 (Remarks to the Author):

Authors addressed most of my concerns or provided reasonable explanations why they were unable to address them. Following my comment, the authors made an attempt to identify SAM-AMP binding site in the CorA model structure. They claim that it is provided in the Extended Data Fig.9 but in the 450734_1_supp_4185642_rxqbw.d.docx file for reviewers I find the Supplemental Fig. 9. a, and b, HPLC image sources of Extended Data Fig. 9a and b, respectively. I find the model data in 450734_1_extended_data_4185653_rxqc4x file.

While it is good to see the structural model and proposed SAM-AMP binding site the conclusion (lines 199-201) that compromised immunity of the R152/R153 and D219/D220 mutant variants in the plasmid challenge assay is consistent with a role in SAM-AMP binding is overreaching since the observed phenotype could be due to the compromised protein folding, oligomer assembly, etc.